# Loss of TRIM29 mitigates viral myocarditis by attenuating PERK-driven ER stress response in male mice

Junying Wang [1,4], Wenting Lu [1,4], Jerry Zhang [1,4], Yong Du [1,4], Mingli Fang [1], Ao Zhang [1], Gabriel Sungcad [1], Samantha Chon [1] & Junji Xing [1,2,3] ✉

Viral myocarditis, an inflammatory disease of the myocardium, is a significant cause of sudden death in children and young adults. The current coronavirus disease 19 pandemic emphasizes the need to understand the pathogenesis mechanisms and potential treatment strategies for viral myocarditis. Here, we found that TRIM29 was highly induced by cardiotropic viruses and promoted protein kinase RNA-like endoplasmic reticulum kinase (PERK)-mediated endoplasmic reticulum (ER) stress, apoptosis, and reactive oxygen species (ROS) responses that promote viral replication in cardiomyocytes in vitro. TRIM29 deficiency protected mice from viral myocarditis by promoting cardiac antiviral functions and reducing PERK-mediated inflammation and immunosuppressive monocytic myeloid-derived suppressor cells (mMDSC) in vivo. Mechanistically, TRIM29 interacted with PERK to promote SUMOylation of PERK to maintain its stability, thereby promoting PERK-mediated signaling pathways. Finally, we demonstrated that the PERK inhibitor GSK2656157 mitigated viral myocarditis by disrupting the TRIM29-PERK connection, thereby bolstering cardiac function, enhancing cardiac antiviral responses, and curbing inflammation and immunosuppressive mMDSC in vivo. Our findings offer insight into how cardiotropic viruses exploit TRIM29-regulated PERK signaling pathways to instigate viral myocarditis, suggesting that targeting the TRIM29-PERK axis could mitigate disease severity.

Myocarditis is clinically and pathologically defined as an inflammatory disease of the myocardium and contributes up to 20% of all sudden deaths in infants, adolescents, and young adults annually[1-3]. Viral infections are a significant cause of myocarditis. Examples include enterovirus coxsackievirus B3 (CVB3), encephalomyocarditis virus (EMCV), influenza virus, adenovirus, parvovirus B19 and Epstein Barr virus[4-6]. Importantly, recent evidence suggests that severe acute respiratory syndrome coronavirus 2 (SARS-CoV-2),

which causes the coronavirus disease 19 (COVID-19) pandemic, is a significant infectious agent for myocarditis[7-10]. Viral myocarditis is a triphasic disease: in phase I, the disease is triggered by viral infection in the heart; in phase II, immune cells infiltrate into the heart and induce uncontrolled antiviral immune and autoimmune responses damaging cardiomyocytes; and in phase III, cardiac remodeling leads to dilated cardiomyopathy[11-15]. However, the precise pathophysiological mechanisms in humans remain elusive,

[1]Department of Surgery and Immunobiology and Transplant Science Center, Houston Methodist Research Institute, Houston Methodist, Houston, TX 77030, USA. [2]Department of Cardiovascular Sciences, Houston Methodist Research Institute, Houston, TX 77030, USA. [3]Department of Surgery, Weill Cornell Medicine, Cornell University, New York, NY 10065, USA. [4]These authors contributed equally: Junying Wang, Wenting Lu, Jerry Zhang, Yong Du. ✉e-mail: jxing@houstonmethodist.org

and we are still in search of effective therapeutic strategies for myocarditis[16,17]. Given the backdrop of the persistent COVID-19 pandemic, there is an urgent need to unravel the intricacies of viral myocarditis' pathogenesis, paving the way for effective therapeutic interventions.

Endoplasmic reticulum (ER) stress arises from the accumulation of unfolded or misfolded proteins in the ER lumen[18]. In mammalian cells, the detection of this stress is primarily facilitated by three ER transmembrane proteins: inositol-requiring enzyme 1α (IRE1α), protein kinase RNA-like ER kinase (PERK; also known as EIF2AK3) and activating transcription factor 6 (ATF6)[18–20]. PERK is the second major branch of the ER stress response and is a protein kinase composed of cytoplasmic and kinase domains[21]. Its cytoplasmic domain senses the accumulation of unfolded/misfolded proteins in the ER lumen[21]. Once activated, PERK undergoes autophosphorylation in its kinase domain and then phosphorylates eukaryotic translation initiation factor 2α (eIF2α)[21]. This sequence of events triggers the activation of the downstream activating transcription factor 4 (ATF4), which in turn upregulates the transcription of C/EBP-homologous protein (CHOP) to induce apoptosis under sustained ER stress[22]. Furthermore, chronic ER stress boosts the generation of reactive oxygen species (ROS) specifically at the ER[23,24]. Notably, PERK plays a pivotal role in facilitating ROS generation when prompted by hydrogen peroxide[25]. Such ER stress is intricately linked to the development of cardiovascular diseases[19,26]. While ROS can serve both protective and harmful roles in cellular processes, their overproduction is observed in conditions like myocarditis[27]. However, the regulation of ER stress factors, especially focusing on PERK and its downstream pathways in the context of myocarditis, is not entirely understood. Similarly, the exact role of ROS in triggering viral myocarditis and its associated mechanisms remain subjects of ongoing investigation.

The prevalent cardiovascular complications associated with COVID-19 underscore the urgent necessity to probe the pathogenic mechanisms of viral myocarditis and identify potential therapeutic interventions. In this study, we observed a significant induction of TRIM29 by cardiotropic viruses. This induction facilitates the recruitment of PERK by TRIM29, promoting its SUMOylation, which in turn ensures its stability within cardiomyocytes. As a result, this process amplifies PERK-mediated pathways leading to ER stress, apoptosis, ROS production, and immunosuppressive monocytic myeloid-derived suppressor cells (mMDSC), fostering the progression of viral myocarditis both in vitro and in vivo. Our work offers compelling in vivo evidence spotlighting the pivotal role of the TRIM29-driven PERK signaling pathway in the pathogenesis of viral myocarditis. Moreover, our findings suggest that the TRIM29-PERK axis holds promise as a viable therapeutic target for managing viral myocarditis.

## Results

### TRIM29 promotes PERK-mediated ER stress and apoptosis induced by cardiotropic viruses in cardiomyocytes

ER stress plays critical roles in the pathogenesis of cardiovascular diseases, including cardiomyopathy and heart failure[19]. Viral myocarditis, characterized by an inflammation of the heart muscle, is predominantly attributed to infections by cardiotropic viruses[4,5,9], and patients with viral myocarditis can progress to cardiomyopathy and heart failure[17,28,29]. To discern the potential correlation between TRIM29 expression, ER stress, and the pathogenesis of viral myocarditis, we initiated an evaluation of TRIM29 levels in patients diagnosed with cardiomyopathy and in mice model with viral myocarditis. We found that both TRIM29 protein and ER stress were dramatically increased in the hearts of mice with cardiotropic virus CVB3-induced myocarditis (Fig. 1a). Parallel findings human subjects indicated

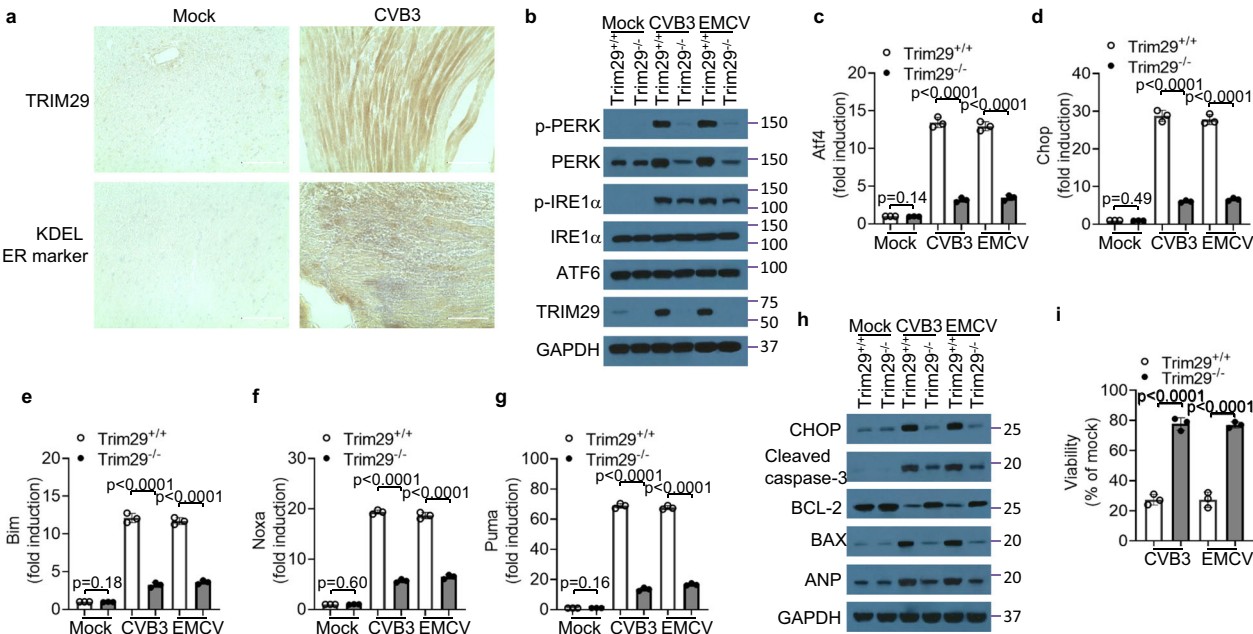

**Fig. 1 | TRIM29 knockout inhibits PERK-mediated ER stress and apoptosis induced by cardiotropic viruses in mouse cardiomyocytes.**
**a** Immunohistochemistry (IHC) analysis of TRIM29 and ER marker KDEL expression in mouse heart tissues from wild-type mice without or with CVB3 infection for 2 days. Scale bars represent 100 μm. **b** Immunoblot analysis of PERK, IRE1α, ATF6 and TRIM29 levels in mouse primary neonatal cardiomyocytes from *Trim29*⁺/⁺ and *Trim29*⁻/⁻ mice by infection without or with CVB3 and EMCV for 6 h at an MOI of 1. RT-qPCR analysis of Atf4 (**c**), Chop (**d**), Bim (**e**), Noxa (**f**) and Puma (**g**) at the mRNA level in mouse primary neonatal cardiomyocytes from *Trim29*⁺/⁺ and *Trim29*⁻/⁻ mice by infection without or with CVB3 and EMCV for 3 h at an MOI of 1. **h** Immunoblot

analysis of CHOP, cleaved caspase-3, BCL-2, BAX and ANP levels in mouse primary neonatal cardiomyocytes from *Trim29*⁺/⁺ and *Trim29*⁻/⁻ mice by infection without or with CVB3 and EMCV for 6 h at an MOI of 1. **i** Cell viability quantification analysis of mouse primary neonatal cardiomyocytes from *Trim29*⁺/⁺ and *Trim29*⁻/⁻ mice by infection without or with CVB3 and EMCV for 12 h at an MOI of 1 using CellTiter-Glo assay. Mock, mouse neonatal cardiomyocytes without infection. Data are shown as the mean ± SD. Statistical significance was determined by a two-tailed, unpaired Student's *t* test. NS, not significant. Data are representative of three independent experiments. Source data are provided as a Source Data file.

heightened TRIM29 and ER stress levels in patients with cardiomyopathy relative to healthy normal controls, as indicated by the TRIM29 and ER stress marker KDEL (Supplementary Fig. 1a). In the context of mammalian cells, there are three ER stress sensor proteins: PERK, IRE1α and ATF6[18–20]. To pinpoint which among these sensor-mediated signaling pathways intersects with TRIM29 expression during viral myocarditis, we assessed the expression levels of PERK, IRE1α and ATF6 in cardiomyocytes. Notably, post-infection with cardiotropic viruses CVB3 and EMCV, AC16 human cardiomyocytes exhibited activated phosphorylation of both PERK and IRE1α (Supplementary Fig. 1b). Concurrently, a significant surge in TRIM29 expression was recorded in both human (Supplementary Fig. 1b) and mouse (Fig. 1b) cardiomyocytes. To delve deeper into TRIM29's regulatory impact on the PERK- or IRE1α-mediated signaling cascades, we adopted a TRIM29 knockdown and knockout approach in cardiomyocytes. Intriguingly, the knockdown of TRIM29 markedly diminished the expression and activation of PERK but not IRE1α in human cardiomyocytes post-infection with the cardiotropic viruses CVB3 and EMCV (Supplementary Fig. 1c). An analogous trend was observed in mouse cardiomyocytes, where TRIM29 knockout led to a drastic reduction in PERK's expression and activation without influencing IRE1α (Fig. 1b). These data suggest that TRIM29 accentuates PERK-mediated ER stress in both human and mouse cardiomyocytes infected with the cardiotropic viruses CVB3 and EMCV.

PERK is known to critically influence the onset of apoptosis[30,31,] and the PERK/ATF4/CHOP signaling cascade is established as a central mechanism driving cell apoptosis, evident both in vitro and in vivo[32,33]. Next, we investigated whether the PERK-mediated ATF4 and CHOP signaling pathways were activated in cardiomyocytes upon infection with cardiotropic viruses. We found that infection with the cardiotropic viruses CVB3 and EMCV substantially increased the mRNA expression of the PERK-regulated transcription factors Atf4 (Fig. 1c, Supplementary Fig. 1d) and Chop (Fig. 1d, Supplementary Fig. 1e) and significantly upregulated the downstream apoptosis-associated regulators Bim (Fig. 1e, Supplementary Fig. 1f), Noxa (Fig. 1f, Supplementary Fig. 1g) and Puma (Fig. 1g, Supplementary Fig. 1h) in both human and mouse cardiomyocytes, whereas TRIM29 knockdown (Supplementary Fig. 1d–h) or knockout (Fig. 1c–g) significantly reduced the expression of the elevated transcription factors Atf4 and Chop and their downstream apoptosis-associated regulators Bim, Noxa and Puma. CHOP is integral to ER stress-induced apoptosis and has been shown to modulate proapoptotic pathways through downregulation of the antiapoptotic protein BCL-2[34] and upregulation of the proapoptotic protein BAX[35]. We proceeded to evaluate if CHOP-mediated apoptosis is triggered in cardiomyocytes upon infection with cardiotropic viruses. We found that infection with cardiotropic viruses significantly enhanced the expression of CHOP, cleaved caspase-3, the proapoptotic protein BAX and the heart failure marker atrial natriuretic peptide (ANP) while reducing the expression of the antiapoptotic protein BCL-2 in both human (Supplementary Fig. 1i) and mouse (Fig. 1h) cardiomyocytes. In contrast, TRIM29 knockdown (Supplementary Fig. 1i) or knockout (Fig. 1h) significantly reduced the expression of CHOP, cleaved caspase-3, BAX and ANP and increased the expression of the antiapoptotic protein BCL-2 in both human and mouse cardiomyocytes infected with CVB3 and EMCV. In line with these findings, infection with the cardiotropic viruses CVB3 and EMCV dramatically reduced the viability of both human (Supplementary Fig. 1j) and mouse (Fig. 1i) cardiomyocytes, whereas TRIM29 knockdown or knockout significantly rescued the viability of cardiomyocytes infected by CVB3 and EMCV (Supplementary Fig. 1j, Fig. 1i). Collectively, these results demonstrate that TRIM29 is highly induced and accentuates PERK-mediated ER stress and apoptosis in cardiomyocytes, subsequent to cardiotropic virus infections in both human and mouse models.

## TRIM29 deficiency unleashes type I interferon production to restrict cardiotropic viruses by relieving ROS-mediated TBK1 inhibition in cardiomyocytes

Persistent ER stress has been linked to an increased production of reactive oxygen species (ROS) within the ER[24]. Next, we investigated whether the ROS response was activated in cardiomyocytes upon infection with cardiotropic viruses. Following exposure to cardiotropic viruses CVB3 and EMCV, we observed a pronounced elevation in ROS levels in both human (Supplementary Fig. 2a) and mouse cardiomyocytes (Fig. 2a). This ROS surge was substantially mitigated by either the knockdown or knockout of TRIM29 (Supplementary Fig. 2a, Fig. 2a). Research indicates that ROS can oxidize the stimulator of interferon genes (STING), inhibiting the downstream activation of TANK-binding kinase 1 (TBK1) and consequently reducing type I interferon (IFN) production[36]. We sought to determine if the heightened ROS levels would oxidize the free thiol groups present in key adaptors of the RNA sensing pathway in cardiomyocytes after infection with cardiotropic viruses. Infection with the cardiotropic viruses CVB3 and EMCV markedly increased oxidation of the free thiol group on TBK1 in both human (Supplementary Fig. 2b) and mouse (Fig. 2b) cardiomyocytes. However, TRIM29 knockdown (Supplementary Fig. 2b) or knockout (Fig. 2b) dramatically reduced the TBK1 oxidization triggered by CVB3 and EMCV in human and mouse cardiomyocytes. Additionally, ROS oxidization of TBK1 triggered by the cardiotropic viruses CVB3 and EMCV blocked TBK1 dimerization and downstream phosphorylation and activation of TBK1 and IRF3 in human (Supplementary Fig. 2c) and mouse (Fig. 2c) cardiomyocytes, whereas TRIM29 knockdown (Supplementary Fig. 2c) or knockout (Fig. 2c) significantly rescued TBK1 dimerization and downstream activation of TBK1 and IRF3. Finally, compared with the shRNA control group or wild-type Trim29+/+ group, TRIM29 knockdown or knockout significantly enhanced the production of IFN-α (Supplementary Fig. 2d, Fig. 2d) and IFN-β (Supplementary Fig. 2e, Fig. 2e), which further restricted the replication of the cardiotropic viruses CVB3 and EMCV (Supplementary Fig. 2f, Fig. 2f) in both human and mouse cardiomyocytes. Taken together, these data suggest that TRIM29 deficiency unleashes type I IFN production to restrict cardiotropic viruses by relieving TBK1 inhibition mediated by ROS in cardiomyocytes.

## Cardiomyocyte-specific TRIM29 deficiency protects mice from viral myocarditis in vivo

Viral myocarditis is a leading cause of congestive heart failure, with CVB3 infection being a primary culprit[17,37]. Since TRIM29 deficiency restricts cardiotropic viruses by reducing PERK-mediated ER stress and apoptosis and unleashing ROS-mediated type I IFN production in vitro, we delved into the role of TRIM29 in controlling the pathogenesis of viral myocarditis in vivo using a CVB3-induced myocarditis mouse model. When subjected to intraperitoneal CVB3 infection, the majority of Trim29+/+ mice exhibited severe susceptibility, while their Trim29−/− counterparts displayed notably improved survival rates (Supplementary Fig. 3a). The heart and pancreas histopathology revealed that Trim29−/− mice had significantly reduced inflammation and infiltration of inflammatory cells in heart or pancreas compared with Trim29+/+ mice following CVB3 infection (Supplementary Fig. 3b). In agreement, echocardiography of Trim29+/+ mice revealed impaired cardiac function (Supplementary Fig. 3c), as evidenced by decreased ejection fraction (EF) (Supplementary Fig. 3d) and fractional shortening (FS) (Supplementary Fig. 3e) compared with those of Trim29/- mice. Compared to Trim29+/+ mice, Trim29/- mice had less heart weight increase during viral myocarditis (Supplementary Fig. 3f), a marker of cardiac inflammatory edema. We also investigated circulating signs of cardiac damage in mice by measuring secretion of the heart-specific isoenzyme of creatine kinase, the biomarker of myocardial cell injury, into the blood and found that creatine kinase was dramatically reduced in

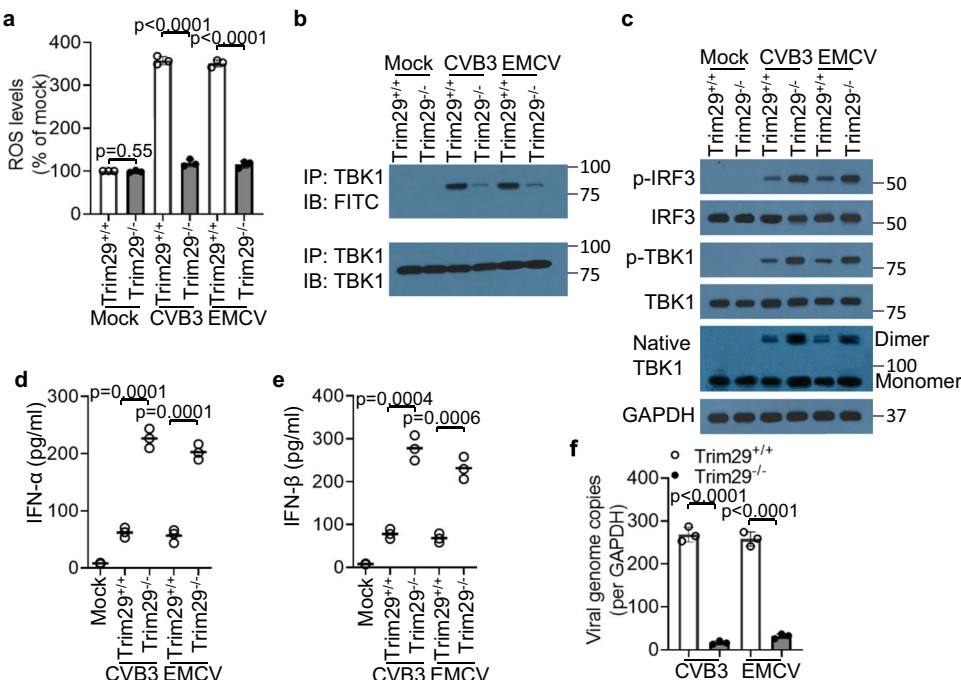

**Fig. 2 | TRIM29 knockout unleashes type I interferon production to restrict cardiotropic viruses by relieving ROS-mediated TBK1 inhibition in mouse cardiomyocytes. a** ROS production analysis of mouse primary neonatal cardiomyocytes from *Trim29*[+/+] and *Trim29*[−/−] mice by infection without or with CVB3 and EMCV for 6 h at an MOI of 1 using a DCFDA cellular ROS assay kit. The fluorescence intensity of the mock group was defined as 100%. **b** Immunoblot (IB) analysis of TBK1 and FITC precipitated with anti-TBK1 from whole-cell lysates incubated with 5 μM 5-IAF for 1 h labeling of free thiols in mouse neonatal cardiomyocytes from *Trim29*[+/+] and *Trim29*[−/−] mice by infection without or with CVB3 and EMCV for 6 h at an MOI of 5. **c** Immunoblot analysis of IRF3, TBK1 and native TBK1 monomer and dimer in mouse neonatal cardiomyocytes from *Trim29*[+/+] and *Trim29*[−/−] mice by infection without or with CVB3 and EMCV for 6 h at an MOI of 5. ELISA of IFN-α (**d**)

and IFN-β (**e**) production by mouse neonatal cardiomyocytes from *Trim29*[+/+] and *Trim29*[−/−] mice by infection without or with CVB3 and EMCV for 16 h at an MOI of 5. Each circle represents one value of the three biological replicates; small horizontal lines indicate the average of triplicates. **f** Quantification of the expression of CVB3 and EMCV viral genome copies relative to GAPDH in mouse neonatal cardiomyocytes from *Trim29*[+/+] and *Trim29*[−/−] mice by infection without or with CVB3 and EMCV for 6 h at an MOI of 1. Mock, mouse neonatal cardiomyocytes without infection. Data are shown as the mean ± SD. Statistical significance was determined by a two-tailed, unpaired Student's *t* test. NS, not significant. Data are representative of three independent experiments. Source data are provided as a Source Data file.

*Trim29*[−/−] mice compared to *Trim29*[+/+] mice (Supplementary Fig. 3g). To further investigate the mechanisms by which *Trim29*[−/−] mice exhibited reduced CVB3-induced myocarditis, we checked type I IFN protein levels in heart homogenates and assessed viral replication in heart, pancreas and spleen tissues by ELISA and plaque-forming assay, respectively. We found that the CVB3 viral loads were significantly reduced in heart, pancreas and spleen from *Trim29*[−/−] mice compared with those from *Trim29*[+/+] mice on day 2 after CVB3 infection (Supplementary Fig. 3h). Furthermore, *Trim29*[−/−] mice had higher concentrations of IFN-α (Supplementary Fig. 3i) and IFN-β (Supplementary Fig. 3j) in the heart than their *Trim29*[+/+] littermates after infection with CVB3. Prior studies have suggested that PERK signaling accentuates the production of proinflammatory cytokines like IL-6, TNF-α and IL-1β[38,39]. In alignment with this, *Trim29*[+/+] mice had higher levels of the cardiac inflammatory cytokines IL-6 (Supplementary Fig. 3k), TNF-α (Supplementary Fig. 3l) and IL-1β (Supplementary Fig. 3m) than their *Trim29*[−/−] littermates after CVB3 infection. These data indicate that complete ablation of TRIM29 protects mice from CVB3-induced myocarditis by enhancing antiviral innate immunity, attenuating inflammation and cardiac damage, and improving cardiac function.

To further investigate the role of cardiomyocyte-specific TRIM29 in controlling the pathogenesis of viral myocarditis in vivo, we crossed *Trim29*-targetd mice with FRT deleter (Rosa26-FLPe) mice to generate *Trim29*-flox mice (*Trim29*[fl/fl]), which were further crossed with *αMyHC- Cre* (WTMyHC-Cre) transgenic mice to generate cardiomyocyte-specific *Trim29* knockout mice, *Trim29*[fl/fl];*αMyHC-Cre* (*Trim29*[MyHC-KO]) (Supplementary Fig. 4a). In addition, the deletion of

*Trim29* was confirmed by PCR analysis of genomic DNA (Supplementary Fig. 4b). Furthermore, mouse primary cardiomyocytes were isolated from wild-type *Trim29*[fl/fl] and *Trim29*[MyHC-KO] mice, and immunoblot analysis confirmed that TRIM29 was deleted in mouse cardiomyocytes from *Trim29*[MyHC-KO] mice infected with CVB3 (Supplementary Fig. 4c). Next, we intraperitoneally infected *Trim29*[fl/fl], WTMyHC-Cre and *Trim29*[MyHC-KO] mice with the cardiotropic virus CVB3 and further evaluated the importance of cardiomyocyte-specific TRIM29 expression in controlling the pathogenesis of viral myocarditis in vivo. Notably, *Trim29*[MyHC-KO] mice manifested a markedly enhanced survival rate compared to their *Trim29*[fl/fl] (Fig. 3a) or WTMyHC-Cre counterparts (Supplementary Fig. 5a). The histopathology revealed that the cardiac inflammation and infiltration of inflammatory cells in *Trim29*[MyHC-KO] hearts were much lower than those in *Trim29*[fl/fl] hearts after CVB3 infection (Fig. 3b). Furthermore, echocardiography of *Trim29*[fl/fl] mice revealed impaired cardiac function (Fig. 3c), as evidenced by decreased ejection fraction (EF) (Fig. 3d) and fractional shortening (FS) (Fig. 3e) compared with *Trim29*[MyHC-KO] mice. Compared to that of *Trim29*[fl/fl] mice, the heart weight gain of *Trim29*[MyHC-KO] mice was dramatically reduced during viral myocarditis (Fig. 3f). Additionally, creatine kinase was dramatically reduced in the circulating blood of *Trim29*[MyHC-KO] mice compared to that of *Trim29*[fl/fl] mice (Fig. 3g). In addition, we found that the CVB3 viral loads were significantly reduced in heart, pancreas and spleen from *Trim29*[MyHC-KO] mice compared with those from *Trim29*[fl/fl] (Fig. 3h) or WTMyHC-Cre (Supplementary Fig. 5b) mice on day 2 following CVB3 infection. Furthermore, *Trim29*[MyHC-KO] mice had

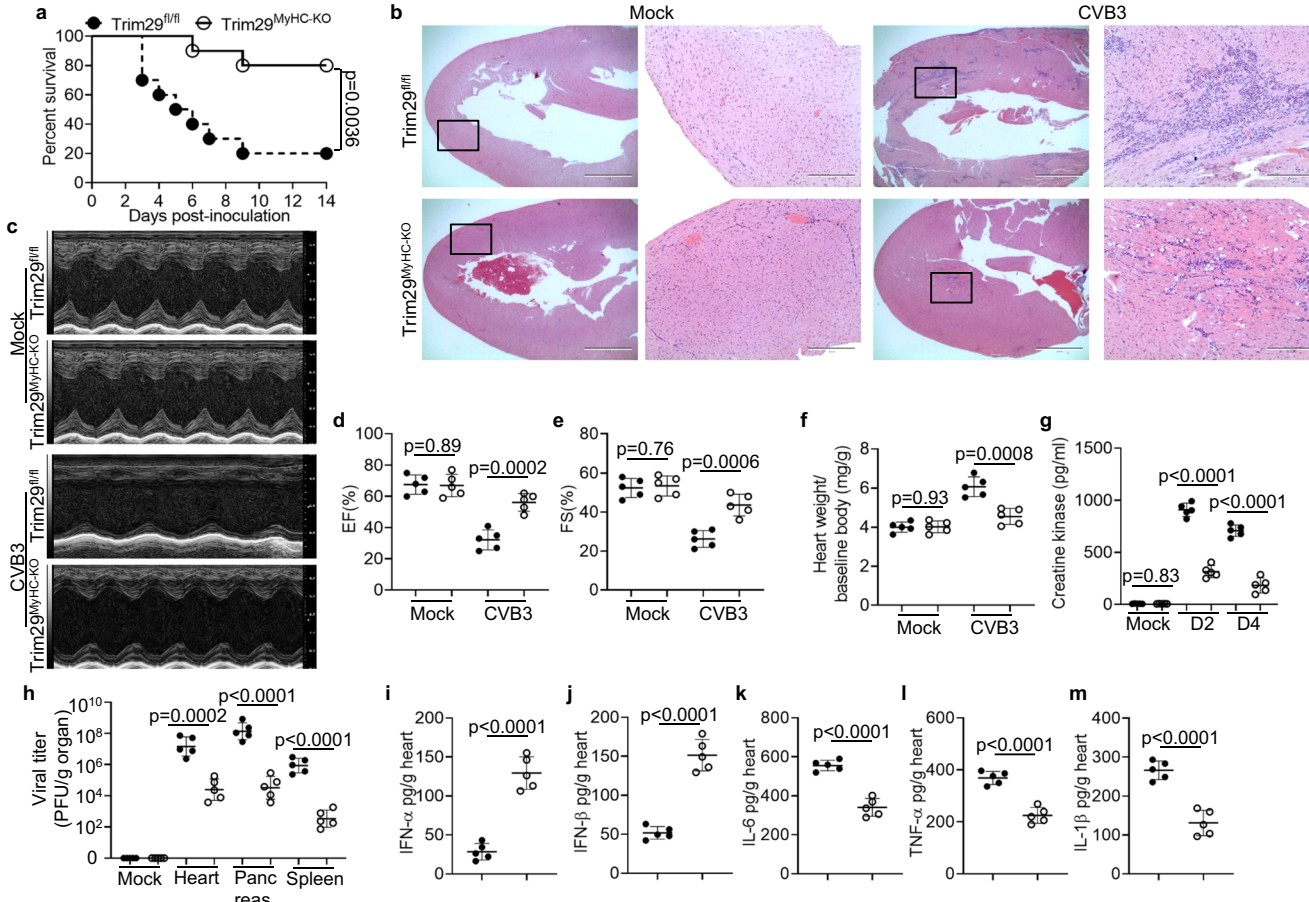

**Fig. 3 | Cardiomyocyte-specific TRIM29 deficiency protects mice from viral myocarditis in vivo. a** Survival of $Trim29^{fl/fl}$ and cardiomyocyte-specific TRIM29 knockout ($Trim29^{MyHC-KO}$) mice after intraperitoneal infection with CVB3 ($1 \times 10^7$ PFU per mouse) ($n = 10$ per group). **b** Hematoxylin and eosin (H&E)-staining of heart sections from $Trim29^{fl/fl}$ and $Trim29^{MyHC-KO}$ mice after intraperitoneal infection without (Mock) or with CVB3 ($1 \times 10^7$ PFU per mouse) for 4 days. Scale bars represent 1000 μm for original images and 400 μm for enlarged images. **c** Representative M-mode echocardiography images of hearts from $Trim29^{fl/fl}$ and $Trim29^{MyHC-KO}$ mice on day 4 after CVB3 infection. Cardiac function analysis of ejection fraction (EF) (**d**) and fractional shortening (FS) (**e**) of hearts from mice as in (**c**) ($n = 5$ per group). **f** Assessment of heart weight/baseline body weight in $Trim29^{fl/fl}$

and $Trim29^{MyHC-KO}$ mice ($n = 5$ per group) on day 0 or day 6 after CVB3 infection. ELISA of creatine kinase production in sera (**g**) and viral titers in homogenates of heart, pancreas, and spleen (**h**) from $Trim29^{fl/fl}$ and $Trim29^{MyHC-KO}$ mice on day 0 (Mock) and day 2 after CVB3 infection ($n = 5$ per group). ELISA of IFN-α (**i**), IFN-β (**j**) IL-6 (**k**), TNF-α (**l**) and IL-1β (**m**) in hearts from $Trim29^{fl/fl}$ and $Trim29^{MyHC-KO}$ mice on day 2 after CVB3 infection ($n = 5$ per group). Data are shown as the mean ± SD. Statistical significance was determined by a two-tailed, unpaired Student's $t$ test and Gehan-Breslow-Wilcoxon test for survival analysis. NS, not significant. Data are representative of three independent experiments. Source data are provided as a Source Data file.

higher concentrations of IFN-α (Fig. 3i, Supplementary Fig. 5c) and IFN-β (Fig. 3j, Supplementary Fig. 5d) in the heart than their $Trim29^{fl/fl}$ or WTMyHC-Cre counterparts after CVB3 infection. In contrast, $Trim29^{fl/fl}$ or WTMyHC-Cre mice had higher levels of the cardiac inflammatory cytokines IL-6 (Fig. 3k, Supplementary Fig. 5e), TNF-α (Fig. 3l, Supplementary Fig. 5f) and IL-1β (Fig. 3m, Supplementary Fig. 5g) than their $Trim29^{MyHC-KO}$ counterparts after CVB3 infection. Finally, we investigated the PERK-mediated ER stress and apoptosis in mouse cardiomyocytes from CVB3 infected $Trim29^{fl/fl}$, WTMyHC-Cre and $Trim29^{MyHC-KO}$ mice. We found that the PERK phosphorylation, the expressions of CHOP, cleaved caspase-3 and the proapoptotic protein BAX were dramatically reduced in mouse cardiomyocytes from CVB3 infected $Trim29^{MyHC-KO}$ mice in comparison to those from $Trim29^{fl/fl}$ or WTMyHC-Cre counterparts (Supplementary Fig. 5h). Collectively, these data indicate that cardiomyocyte-specific TRIM29 deficiency is sufficient to protect mice from CVB3-induced myocarditis by promoting cardiac antiviral function, attenuating inflammation, reducing cardiac ER stress and apoptosis, and improving cardiac function in vivo.

## TRIM29 deficiency reduces PERK-mediated immunosuppressive mMDSC to enhance functions of antiviral CD8 T cells during viral myocarditis in vivo

PERK signaling modulates immunosuppression in tumors by myeloid cells[40] and is integral to MDSC production[41]. Proinflammatory cytokines such as IL-1β and IL-6 can recruit and activate MDSC[42]. Since there is more PERK activation and higher cardiac inflammatory cytokines IL-6 and IL-1β in the hearts of wild-type $Trim29^{fl/fl}$ mice, we investigated whether enhanced PERK, IL-6 and IL-1β mediated more recruitment and activation of MDSC in $Trim29^{fl/fl}$ mice during viral myocarditis by flow cytometry. Flow cytometry analysis revealed an augmented frequency of MDSC (CD11b⁺Gr1⁺ cells) infiltration in the heart (Fig. 4a, b, Supplementary Fig. 6a) and spleen (Fig. 4c, d, Supplementary Fig. 6b) from $Trim29^{fl/fl}$ mice in comparison to $Trim29^{MyHC-KO}$ (Fig. 4a–d) or WTMyHC-Cre (Supplementary Fig. 6a, b) mice after CVB3 infection for 2 days. In contrast, the frequencies of MDSC in spleens from $Trim29^{fl/fl}$ and $Trim29^{MyHC-KO}$ mice were comparable before CVB3 infection (Fig. 4c, d). MDSC are heterogeneous population of immature myeloid cells that include monocytic (mMDSC) and granulocytic (gMDSC)

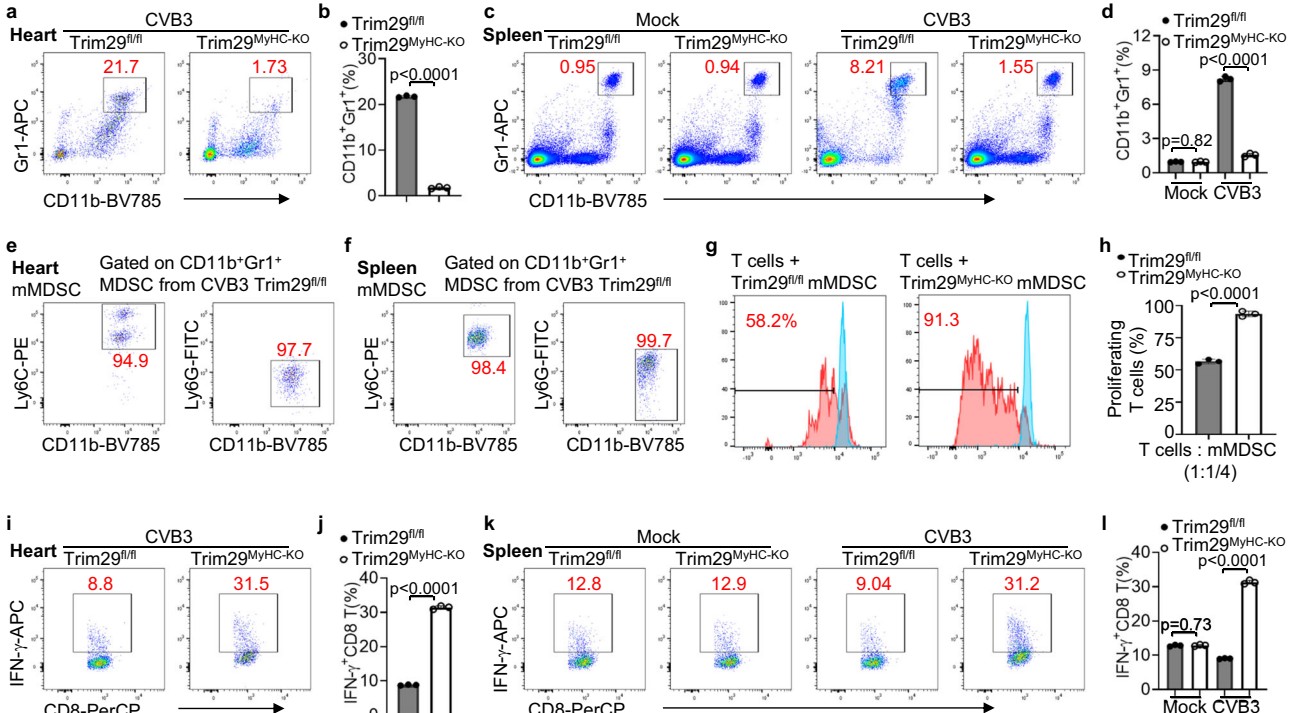

**Fig. 4 | TRIM29 deficiency reduces PERK-mediated immunosuppressive mMDSC to enhance functions of antiviral CD8 T cells during viral myocarditis in vivo.** Flow cytometry (**a**, **c**) and quantification (**b**, **d**) analysis of mouse MDSC (CD11b⁺Gr1⁺) cells of heart infiltrated immune cells (**a**, **b**) and spleen immune cells (**c**, **d**) from *Trim29*^fl/fl^ and *Trim29*^MyHC-KO^ mice infected without or with CVB3 for 2 days using CD11b-BV785 and Gr1-APC antibodies. Flow cytometry analysis of mouse mMDSC in gated MDSC (CD11b⁺Gr1⁺) from heart (**e**) and spleen (**f**) of CVB3 infected *Trim29*^fl/fl^ mice as in (**a**), (**c**) using Ly6C-PE and Ly6G-FITC antibodies. Flow cytometry (**g**) and quantification (**h**) analysis of proliferating CD8 T cells cocultured with mouse MDSC (1:1/4 ratio) isolated from heart of *Trim29*^fl/fl^ and *Trim29*^MHC-KO^ mice with a 3-day infection of CVB3 infection for 0 (blue line) and 3 days (red line)

using carboxyfluroescein succinimidyl ester (CFSE) staining. Flow cytometry (**i**, **k**) and quantification (**j**, **l**) analysis of mouse IFN-γ producing CD8 T cells (IFN-γ⁺ CD8 T cells) of heart infiltrated lymphocytes (**i**, **j**) and spleen lymphocytes (**k**, **l**) from *Trim29*^fl/fl^ and *Trim29*^MyHC-KO^ mice infected without or with CVB3 for 2 days followed by PMA/ionomycin stimulation for 4 h using CD8-PerCP and IFN-γ-APC antibodies. Flow cytometry data were acquired on an LSR-II flow cytometer (Beckton Dickinson) and analyzed using FlowJo v10 software (Tree Star). Data are shown as the mean ± SD. Statistical significance was determined by a two-tailed, unpaired Student's t test. NS, not significant. Data are representative of three independent experiments. Source data are provided as a Source Data file.

subsets[43]. The mMDSCs are characterized by CD11b⁺Ly6C⁺Ly6G⁻ phenotype compared with the gMDSC subset have a CD11b⁺Ly6C⁻Ly6G⁺ phenotype. Next, we investigated the identity of these MDSC subsets in heart and spleen using Ly6C and Ly6G antibodies by flow cytometry. Flow cytometry analysis identified these MDSC (CD11b⁺Gr1⁺ cells) gated in heart (Fig. 4e) and spleen (Fig. 4f) from CVB3 infected *Trim29*^fl/fl^ mice were characterized by CD11b⁺Ly6C⁺Ly6G⁻, suggesting these MDSC in heart and spleen are mMDSC. Additionally, we found there were significantly more cell numbers of mMDSC, macrophages, neutrophils, B cells and T cells infiltrated in heart from *Trim29*^fl/fl^ mice in comparison to *Trim29*^MyHC-KO^ mice (Supplementary Fig. 6c) after CVB3 infection for 2 days. PERK ablation in MDSC elicits antitumor immunity by promoting the proliferation and activation of CD8 T cells[40]. In adaptive immunity, CD8 T cells play an essential role in controlling viral infection by killing virus-infected cells and producing effector cytokines[44,45]. Next, we investigated whether MDSC from *Trim29*^MyHC-KO^ mice elicited functions of antiviral CD8 T cells by assessing the suppression of T cell proliferation by MDSC. We found that MDSC from CVB3-infected hearts of *Trim29*^MyHC-KO^ mice promoted more proliferation of CD8 T cells than MDSC from *Trim29*^fl/fl^ mice (Fig. 4g, h). Additionally, flow cytometry analysis showed in vitro stimulation with phorbol 12-myristate 13-acetate and ionomycin significantly enhanced the IFN-γ-producing ability of CD8 T cells (Fig. 4i–l) in the heart (Fig. 4i, j) and spleen (Fig. 4k, l) from *Trim29*^MyHC-KO^ mice in comparison to *Trim29*^fl/fl^ mice after CVB3 infection for 2 days. In contrast, the IFN-γ-producing ability of CD8 T cells (Fig. 4k, l) in spleens from *Trim29*^fl/fl^ and *Trim29*^MyHC-KO^ mice were comparable before CVB3 infection. Taken

together, these data demonstrate that TRIM29 deficiency reduces PERK-mediated immunosuppressive mMDSC to enhance functions of antiviral CD8 T cells during viral myocarditis in vivo.

## TRIM29 interacts with PERK to promote ER stress and apoptosis to enhance viral replication

Next, we explored the molecular mechanisms by which TRIM29 mitigated PERK-mediated ER stress and ROS responses, thereby accentuating the pathogenesis of viral myocarditis. Initially, we sought to identify potential TRIM29 targets in mouse cardiomyocytes after CVB3 infection. Leveraging an antibody specific to TRIM29, we immunoprecipitated TRIM29-associated proteins from lysates of CVB3-infected mouse cardiomyocytes. Subsequent analysis via liquid chromatography-mass spectrometry identified a spectrum of TRIM29-associated proteins, among which the eukaryotic translation initiation factor 2-alpha kinase 3 (EIF1AK3)-encoded PERK prominently stood out (Supplementary Table 1). We next investigated whether TRIM29 could interact directly with PERK in cardiomyocytes at the endogenous protein level. The anti-PERK antibody, but not control IgG, precipitated TRIM29 in mouse cardiomyocytes from *Trim29*^fl/fl^ or WTMyHC-Cre mice after infection with CVB3 or EMCV (Fig. 5a). However, the interaction between TRIM29 and PERK was disrupted in mouse cardiomyocytes from *Trim29*^fl/fl^ or WTMyHC-Cre mice with mock infection (Fig. 5a). To delineate the specific regions of interaction between TRIM29 and PERK, we analyzed interactions among Myc-tagged recombinant TRIM29 and HA-tagged recombinant full-length PERK, as well as truncation mutants of PERK (Fig. 5b). Both full-length PERK

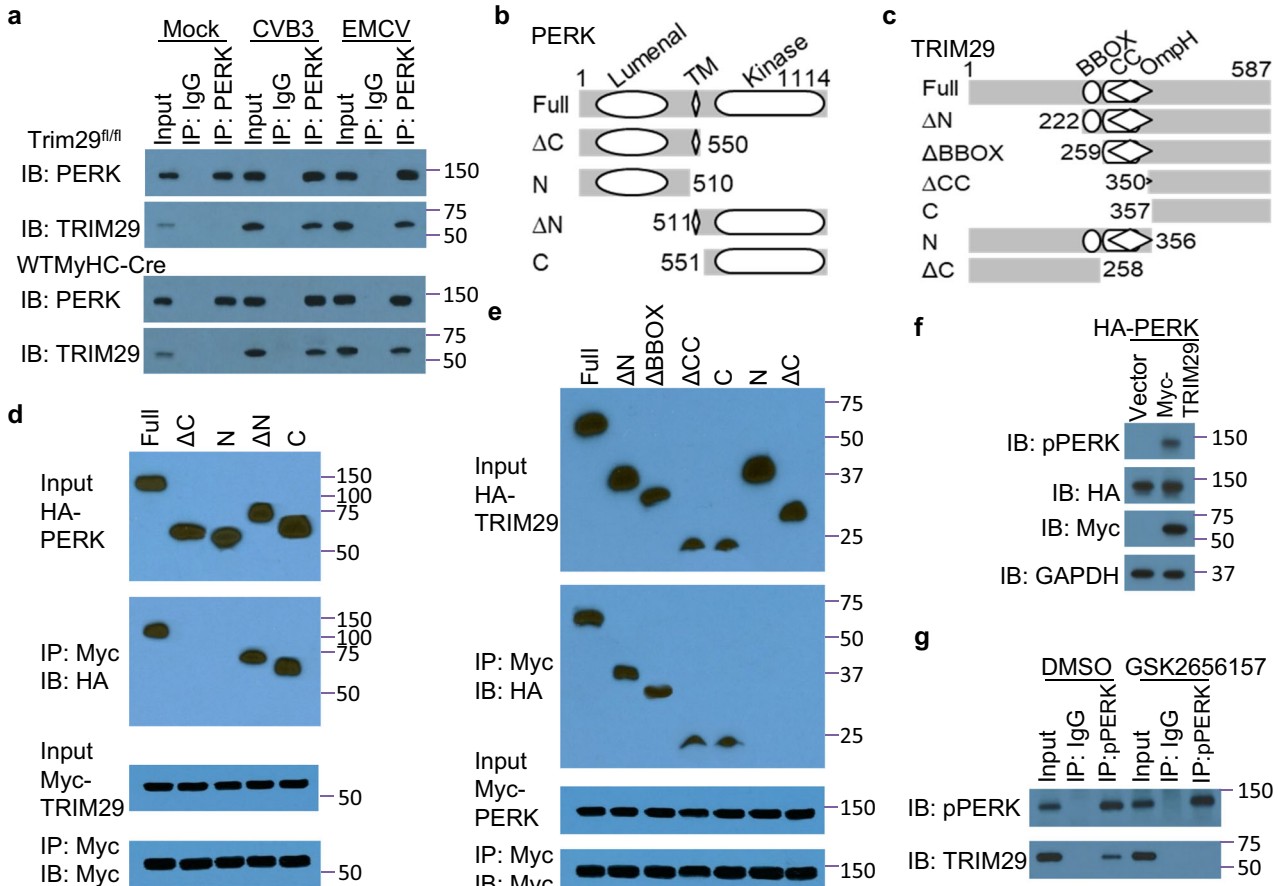

**Fig. 5 | TRIM29 interacts with PERK to promote PERK-mediated ER stress and apoptosis. a** Immunoblot (IB) analysis of endogenous proteins TRIM29 and PERK precipitated with anti-PERK or immunoglobulin G (IgG) from whole-cell lysates of mouse primary neonatal cardiomyocytes from *Trim29*fl/fl and WTMyHC-Cre mice with or without (Mock) CVB3 and EMCV infection for 6 h at an MOI of 5. Schematic diagram showing full-length PERK (Full, **b**) or TRIM29 (Full, **c**) and serial truncations of PERK (**b**) or TRIM29 (**c**) with deletion (Δ) of various domains (left margin); numbers at ends indicate amino acid positions (top). Luminal, ER luminal domain; TM, transmembrane domain; Kinase, protein kinase domain; BBOX, the B-box zinc-finger domain; CC, the coil-coil domain; OmpH, the outer membrane protein H domain. Immunoblot analysis of purified HA-tagged full-length PERK and serial truncations of PERK with deletion of various domains alone (**d**) or purified HA-tagged full-length TRIM29 and serial truncations of TRIM29 with deletion of various domains alone (**e**) with anti-HA antibody (top blot) or after incubation with Myc-tagged TRIM29 (**d**) or Myc-tagged PERK (**e**) and immunoprecipitation with anti-Myc antibody (second blot), and immunoblotting analysis of purified Myc-tagged TRIM29 (**d**) or Myc-tagged PERK (**e**) with anti-Myc antibody (third blot) or after incubation with Myc-tagged TRIM29 (**d**) or Myc-tagged PERK (**e**) and immunoprecipitation with anti-Myc antibody (bottom blot). **f** Immunoblot analysis of endogenous phosphorylated PERK (pPERK), HA-tagged PERK and Myc-tagged TRIM29 from whole-cell lysates of HEK293T cells co-transfected with HA-PERK and vector or Myc-TRIM29 for 24 h. **g** Immunoblot analysis of endogenous proteins TRIM29 and phosphorylated PERK (pPERK) precipitated with anti-pPERK or immunoglobulin G (IgG) from whole-cell lysates of mouse primary neonatal cardiomyocytes from *Trim29*fl/fl mice followed by CVB3 infection at an MOI of 5 and treatment with the PERK inhibitor GSK2656157 or DMSO for 6 h. The position of protein markers (shown in kDa) is indicated on the right. Data are representative of three independent experiments. Source data are provided as a Source Data file.

and the C-terminal protein kinase domain of PERK bound to TRIM29 (Fig. 5d). Additionally, the mapping results for Myc-tagged recombinant PERK and HA-tagged full-length TRIM29 and their truncation mutants (Fig. 5c) showed that the C-terminal domain of TRIM29 bound to PERK (Fig. 5e). Additionally, overexpression of TRIM29 promoted the autophosphorylation and stability of PERK compared with vector control (Fig. 5f). These results indicate that TRIM29 interacts with PERK to promote its autophosphorylation and stability in cardiomyocytes during viral myocarditis.

Furthermore, we investigated whether TRIM29 interacted with PERK to promote ER stress and apoptosis to enhance viral replication using the PERK inhibitor GSK2656157. We evaluated the effect of different concentrations of GSK2656157 on the viability of mouse cardiomyocytes and found that GSK2656157 concentrations of 1 μM and lower did not affect the viability of cardiomyocytes (Supplementary Fig. 7a). Compared with the DMSO control, treatment with the PERK inhibitor GSK2656157 disrupted the interaction of TRIM29 and active

phosphorylated PERK (pPERK) in mouse cardiomyocytes infected with CVB3 (Fig. 5g). Additionally, treatment with the PERK inhibitor GSK2656157 significantly reduced the replication of CVB3 (Supplementary Fig. 7b) and EMCV (Supplementary Fig. 7c) in mouse cardiomyocytes. We also found that treatment with GSK2656157 significantly reduced the PERK-regulated transcription factors Atf4 (Supplementary Fig. 8a) and CHOP (Supplementary Fig. 8b) and the downstream apoptosis-associated regulators Bim (Supplementary Fig. 8c), Noxa (Supplementary Fig. 8d) and Puma (Supplementary Fig. 8e) activated by CVB3 or EMCV infection in mouse cardiomyocytes. Collectively, these data suggest that TRIM29 interacts with PERK to promote ER stress and apoptosis, thereby fostering an environment conducive for enhanced viral replication.

**TRIM29 promotes SUMOylation of PERK to maintain stability**

Considering the observed interaction between TRIM29 and PERK, coupled with heightened PERK expression in wild-type

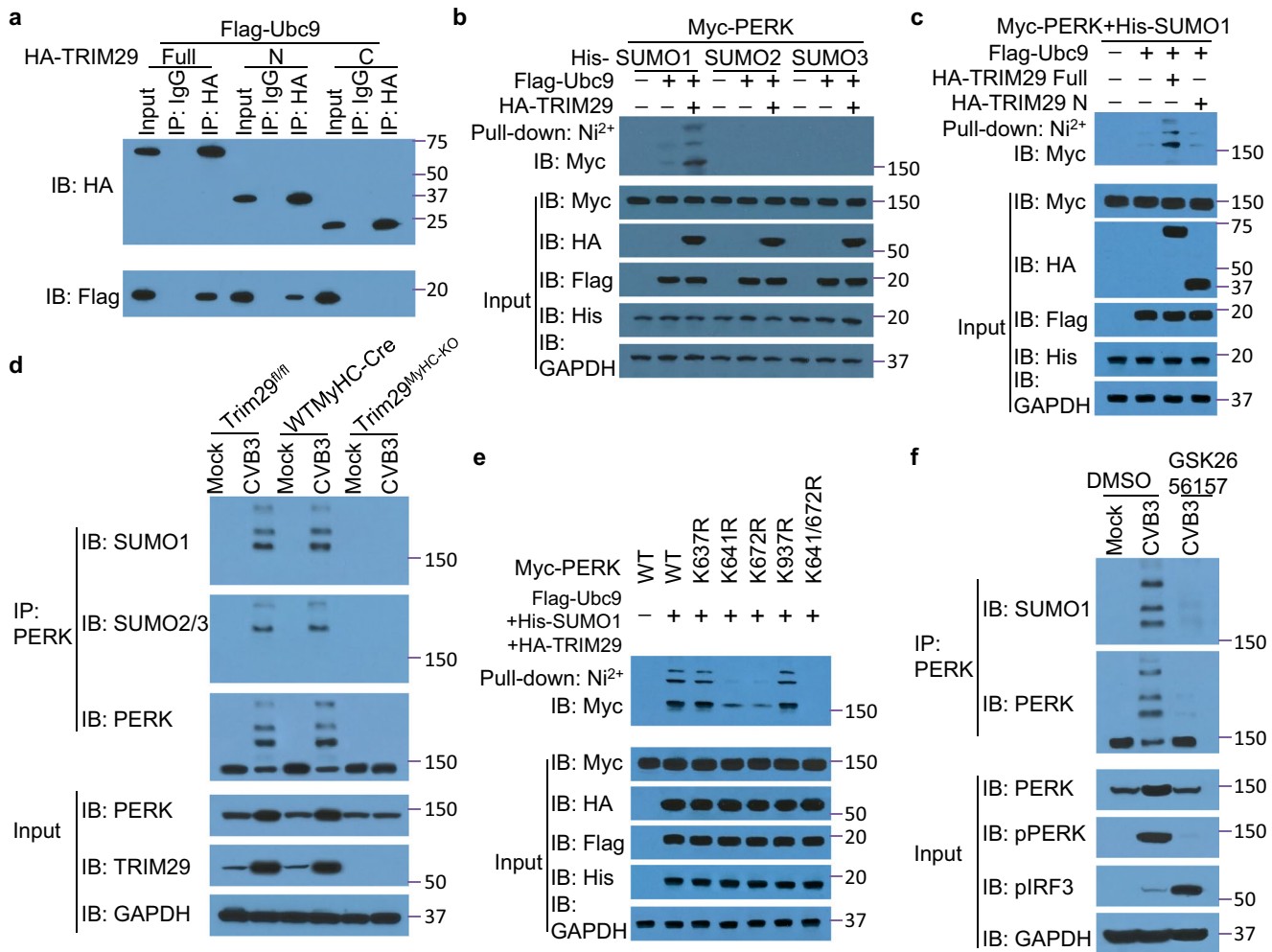

**Fig. 6 | TRIM29 promotes SUMOylation of PERK to maintain its stability.**
**a** Immunoblot (IB) analysis of HA-TRIM29 and Flag-Ubc9 precipitated with anti-HA or immunoglobulin G (IgG) from whole-cell lysates of HEK293T cells transfected with Flag-Ubc9 and HA-TRIM29 Full, its mutants HA-TRIM29 N or C. **b** Immunoblot analysis of TRIM29-mediated PERK SUMOylation with SUMO1, SUMO2 or SUMO3 from whole-cell lysates of HEK293T cells transfected with the indicated plasmids for 24 h followed by Ni$^{2+}$-NTA agarose affinity pull-down assay using anti-HA, anti-Myc, anti-Flag and anti-His antibodies. **c** Immunoblot analysis of SUMO1-mediated PERK SUMOylation by TRIM29 Full or its mutant N from whole-cell lysates of HEK293T cells transfected with the indicated plasmids for 24 h followed by Ni$^{2+}$-NTA agarose affinity pull-down assay using anti-HA, anti-Myc, anti-Flag and anti-His antibodies. **d** Immunoblot analysis of endogenous SUMO1-mediated PERK SUMOylation by TRIM29 precipitated with anti-PERK from whole-cell lysates of mouse primary neonatal cardiomyocytes from *Trim29*$^{fl/fl}$, WTMyHC-Cre and

*Trim29*$^{MyHC-KO}$ mice without (Mock) or with CVB3 infection at an MOI of 5 using anti-PERK, anti-TRIM29, anti-SUMO1 and anti-SUMO2/3 antibodies. **e** Immunoblot analysis of TRIM29-mediated SUMOylation of wild-type (WT) PERK or its mutants, including K637R, K641R, K672R, K937R and K641/672R, from whole-cell lysates of HEK293T cells transfected with the indicated plasmids for 24 h followed by Ni$^{2+}$-NTA agarose affinity pull-down assay using anti-HA, anti-Myc, anti-Flag and anti-His antibodies. **f** Immunoblot analysis of endogenous SUMO1-mediated PERK SUMOylation by TRIM29 precipitated with anti-PERK from whole-cell lysates of mouse primary neonatal cardiomyocytes from *Trim29*$^{fl/fl}$ mice without (Mock) or with CVB3 infection at an MOI of 5 and treatment with the PERK inhibitor GSK2656157 or DMSO using anti-PERK, anti-TRIM29 and anti-SUMO1 antibodies. The position of protein markers (shown in kDa) is indicated on the right. Data are representative of three independent experiments. Source data are provided as a Source Data file.

cardiomyocytes, we delved deeper into the molecular intricacies underlying TRIM29's role in enhancing PERK's expression and stability. SUMOylation is involved in mediating protein–protein interactions and protein stability[46]. Interestingly, our analysis identified the ubiquitin-conjugating enzyme E2I (Ubc9), a SUMO-conjugating enzyme 2 ligase, as a member of the TRIM29-interacting protein cohort (Supplementary Table 1). It has been reported that certain TRIM family members can act as E3 SUMO ligases, which utilize Ubc9 as an E2 ligase[47]. We first examined whether TRIM29 served as an E3 SUMO ligase and recruited the Ubc9 E2 enzyme to induce SUMOylation of PERK. We found that full-length TRIM29 and its N mutant, but not its C mutant, interacted with the SUMO-conjugating E2 enzyme Ubc9 (Fig. 6a). As expected, we found that there were two SUMO Interacting Motifs (SIMs), AA 140–143 and AA 282–288 in TRIM29 mutant N (Supplementary Table 2). Additionally, TRIM29 promoted SUMO1 but

not SUMO2 or SUMO3 modification of PERK (Fig. 6b). In addition, full-length TRIM29, but not its N mutant, promoted SUMO1-mediated SUMOylation of PERK (Fig. 6c). We then determined whether endogenous PERK was SUMOylated by TRIM29 in mouse cardiomyocytes. We observed PERK SUMOylation orchestrated by TRIM29 in CVB3-infected mouse cardiomyocytes from *Trim29*$^{fl/fl}$ and WTMyHC-Cre mice, an effect absent in *Trim29*$^{MyHC-KO}$ mouse cardiomyocytes (Fig. 6d). The PERK SUMOylation was majorly mediated by SUMO1, with a little bit of SUMO2/3. In contrast, in the absence of CVB3 infection, PERK SUMOylation remained undetected in three sets of cardiomyocytes (Fig. 6d). To pinpoint the SUMOylation sites on PERK modulated by TRIM29, we employed the SUMOylation site prediction tool, JASSA[48]. We identified four candidate SUMOylation sites with the best prediction scores and density, including K637, K641, K672 and K937 in murine PERK (Supplementary Table 2), and replaced each of

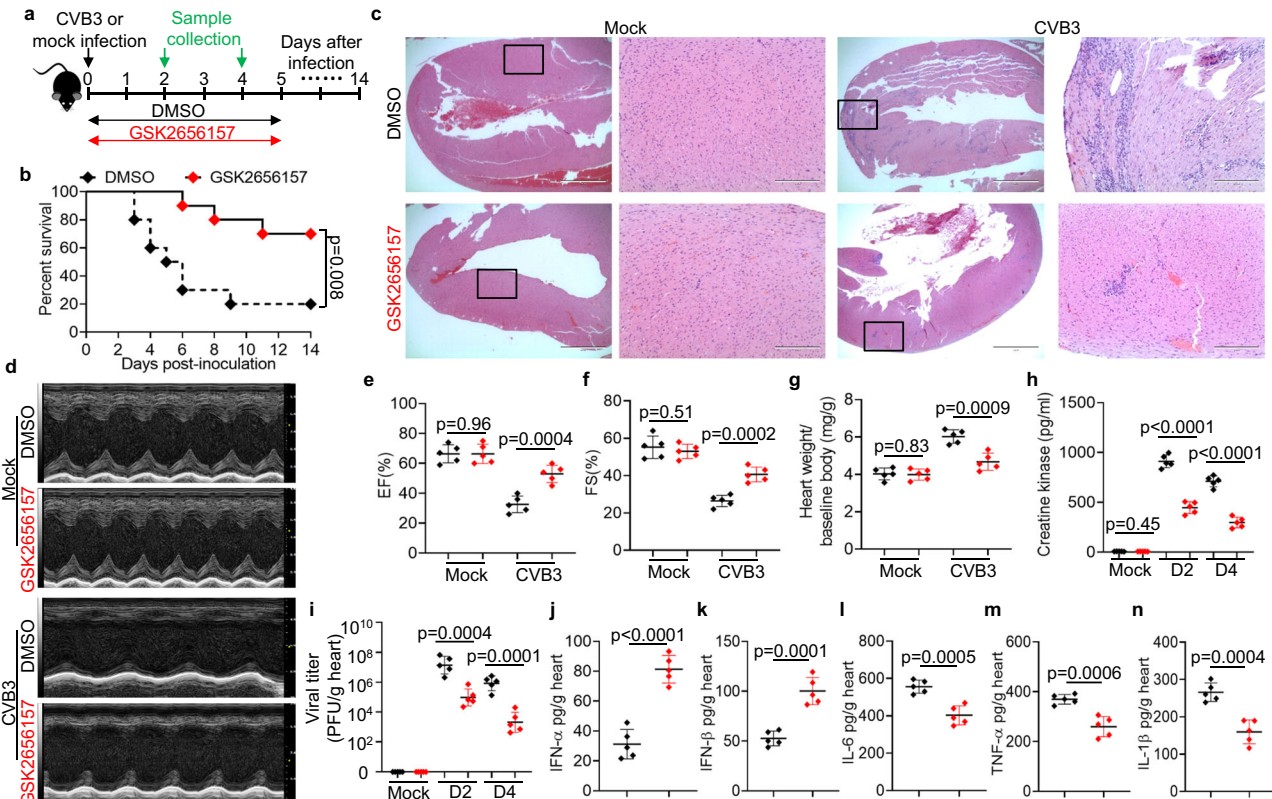

**Fig. 7 | A PERK inhibitor ameliorates viral myocarditis in vivo. a** Schematic illustration of the animal experiment. **b** Survival of wild-type (WT) mice after intraperitoneal infection with CVB3 ($1 \times 10^7$ PFU per mouse) and treatment with the PERK inhibitor GSK2656157 or DMSO ($n = 10$ per group). **c** Hematoxylin and eosin (H&E)-staining of heart sections from WT mice after intraperitoneal infection without (Mock) or with CVB3 ($1 \times 10^7$ PFU per mouse) and treatment with the PERK inhibitor GSK2656157 or DMSO for 4 days. Scale bars represent 1000 μm for original images and 400 μm for enlarged images. **d** Representative M-mode images of hearts from WT mice after intraperitoneal infection without (Mock) or with CVB3 and treatment with the PERK inhibitor GSK2656157 or DMSO for 4 days by echocardiography analysis. Cardiac function analysis of ejection fraction (EF) (**e**) and fractional shortening (FS) (**f**) of hearts from mice as in (**d**) ($n = 5$ per group).

**g** Assessment of heart weight/baseline body weight in WT mice ($n = 5$ per group) on day 0 or day 6 after CVB3 infection and treatment with the PERK inhibitor GSK2656157 or DMSO. ELISA of creatine kinase production in sera (**h**) and viral titers in homogenates of hearts (**i**) from WT mice on day 0 (Mock), day 2 and day 4 after CVB3 infection and treatment with the PERK inhibitor GSK2656157 or DMSO ($n = 5$ per group). ELISA of IFN-α (**j**), IFN-β (**k**) IL-6 (**l**), TNF-α (**m**) and IL-1β (**n**) in hearts from WT mice on day 2 after CVB3 infection and treatment with the PERK inhibitor GSK2656157 or DMSO ($n = 5$ per group). Data are shown as the mean ± SD. Statistical significance was determined by a two-tailed, unpaired Student's $t$ test and Gehan-Breslow-Wilcoxon test for survival analysis. NS, not significant. Data are representative of three independent experiments. Source data are provided as a Source Data file.

those lysine residues individually with arginine. Mutagenesis indicated that mutation of either K641 or K672, but not K637 or K937, dramatically reduced the SUMOylation of PERK (Fig. 6e), while simultaneous mutation of both K641 and K672 (K641/672 R) abolished all the SUMO1 moieties from PERK (Fig. 6e). Compared with the DMSO control, treatment with the PERK inhibitor GSK2656157 almost abolished all the SUMO1 moieties from PERK in mouse cardiomyocytes infected with CVB3 (Fig. 6f). Collectively, these data suggest that TRIM29 promotes SUMOylation of PERK to maintain its stability.

**PERK inhibitor ameliorates viral myocarditis in vivo**
Because the PERK inhibitor GSK2656157 disrupts the interaction of TRIM29 and phosphorylated PERK, reduces PERK-mediated ER stress and apoptosis, abolishes TRIM29-induced PERK SUMOylation and restricts replication of the cardiotropic viruses CVB3 and EMCV in vitro, we next investigated whether the PERK inhibitor GSK2656157 treated viral myocarditis in a wild-type mouse model of CVB3-induced myocarditis in vivo. We first intraperitoneally infected wild-type mice with the cardiotropic virus CVB3 and then injected the mice intraperitoneally with the PERK inhibitor GSK2656157 for 5 days and monitored survival over time (Fig. 7a). While all DMSO-treated mice succumbed to CVB3 infection, the survival of GSK2656157-treated mice was

significantly improved (Fig. 7b). The heart histopathology revealed that GSK2656157-treated mice had significantly reduced cardiac inflammation and infiltration of inflammatory cells compared with their DMSO-treated littermates following CVB3 infection (Fig. 7c). In agreement, echocardiography of DMSO-treated mice revealed impaired cardiac function (Fig. 7d), as evidenced by decreased ejection fraction (EF) (Fig. 7e) and fractional shortening (FS) (Fig. 7f) compared with their GSK2656157-treated littermates. Compared to that of DMSO-treated mice, the heart weight gain of GSK2656157-treated mice was dramatically reduced during viral myocarditis (Fig. 7g). In addition, creatine kinase was dramatically reduced in the circulating blood of GSK2656157-treated mice compared to their DMSO-treated littermates (Fig. 7h). In addition, we found that the CVB3 viral loads were significantly reduced in hearts from GSK2656157-treated mice compared with their DMSO-treated littermates on day 2 and day 4 following CVB3 infection (Fig. 7i). Furthermore, GSK2656157-treated mice had higher concentrations of IFN-α (Fig. 7j) and IFN-β (Fig. 7k) in the heart than their DMSO-treated littermates after CVB3 infection. In contrast, DMSO-treated mice had higher levels of the cardiac inflammatory cytokines IL-6 (Fig. 7l), TNF-α (Fig. 7m) and IL-1β (Fig. 7n) than their GSK2656157-treated littermates after CVB3 infection. These results suggest that treatment with the PERK inhibitor GSK2656157

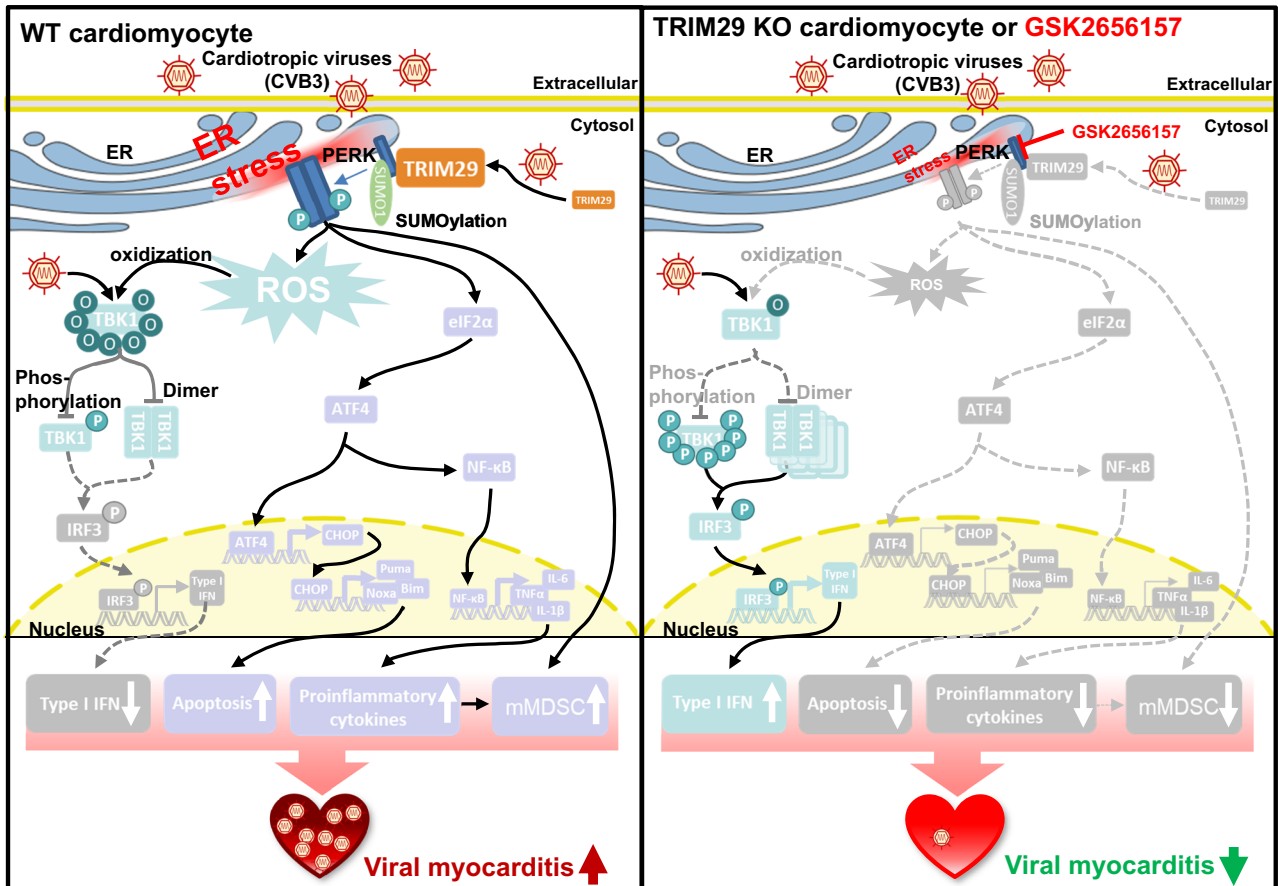

**Fig. 8 | Working model for TRIM29-PERK axis in controlling pathogenesis of viral myocarditis by regulating ER stress and ROS responses.** In WT cardiomyocytes after infection with cardiotropic viruses (CVB3), TRIM29 is strongly induced by cardiotropic viruses and interacts with PERK and induces its SUMO1-mediated SUMOylation for maintaining stability of PERK, thereby promoting ROS-mediated TBK1 oxidization to reduce TBK1-mediated type I IFN production and enhancing PERK ER stress-mediated apoptosis, proinflammatory cytokines, and

immunosuppressive mMDSC, which act in combination to cause pathogenesis of viral myocarditis. In contrast, in TRIM29 KO cardiomyocytes or WT cardiomyocytes with GSK2656157 treatment, PERK could not maintain its stability without TRIM29 or with GSK2656157 treatment post infection of cardiotropic viruses, and loss its strong ability to promote ER stress and ROS responses, resulting in enhanced type I IFN and reduced apoptosis, proinflammatory cytokines and immunosuppressive mMDSC, which significantly reduce viral myocarditis.

ameliorates viral myocarditis by improving cardiac function and enhancing cardiac antiviral innate immunity in vivo.

We next investigated whether treatment with GSK2656157 reduces the immunosuppressive mMDSC activated by PERK-mediated ER stress during viral myocarditis in vivo. Flow cytometry analysis revealed much lower frequencies of mMDSC infiltration in the heart (Supplementary Fig. 9a, b) and spleen (Supplementary Fig. 9c, d) from GSK2656157-treated mice in comparison to their DMSO-treated counterparts after CVB3 infection for 2 days. Additionally, flow cytometry analysis showed in vitro stimulation with phorbol 12-myristate 13-acetate and ionomycin significantly enhanced the IFN-γ-producing ability of CD8 T cells in the heart (Supplementary Fig. 9e, f) and spleen (Supplementary Fig. 9g, h) from GSK2656157-treated mice in contrast to DMSO-treated mice after CVB3 infection for 2 days. Finally, we investigated the PERK-mediated ER stress and apoptosis in mouse cardiomyocytes from CVB3 infected mice treated with DMSO or GSK2656157. We found that the PERK phosphorylation, the expressions of CHOP, cleaved caspase-3 and the proapoptotic protein BAX were dramatically reduced in mouse cardiomyocytes from GSK2656157-treated mice in comparison to their DMSO-treated counterparts after CVB3 infection for 2 days (Supplementary Fig. 9i). Collectively, these data indicate that treatment with the PERK inhibitor GSK2656157 ameliorates viral myocarditis by promoting cardiac antiviral function, attenuating inflammation, reducing cardiac ER stress

and apoptosis, and reducing immunosuppressive mMDSC to enhance functions of antiviral CD8 T cells in vivo.

## Discussion

TRIM29 is a multifunctional protein belonging to the TRIM family of E3 ubiquitin ligases[49] and has been shown to play critical roles in multiple cancers[50] and antiviral innate immunity[51–53]. However, the role of TRIM29 in cardiovascular diseases, such as myocarditis, has remained elusive. In this study, we discovered that TRIM29 was an essential regulator of PERK-mediated signaling pathways for causing pathogenesis of viral myocarditis and that the TRIM29-PERK axis could be targeted to reduce viral myocarditis in a mouse model of CVB3-induced myocarditis in vivo (Fig. 8). Both human and mouse cardiomyocytes, when infected by cardiotropic viruses CVB3 and EMCV, exhibit a pronounced expression of TRIM29. This upregulation augments PERK-mediated ER stress and escalates PERK/ATF4/CHOP-driven apoptosis. Moreover, TRIM29 is observed to bolster ROS levels, which in turn modulates TBK1 through oxidation, inhibiting its function. This cascade ultimately diminishes type I IFN production, thereby favoring the replication of cardiotropic viruses. Corroboratively, our Trim29 total knockout (Trim29[−/−]) and cardiomyocyte-specific Trim29 knockout (Trim29[MyHC-KO]) mouse models displayed resistance to viral myocarditis, which were achieved through improved cardiac function, enhanced cardiac antiviral responses, and curtailed PERK-driven ER

stress, apoptosis and immunosuppressive mMDSC. Delving deeper into the underlying molecular intricacies, our immunoprecipitation-mass spectrometry analyses (Supplementary Table 1) revealed PERK among the suite of TRIM29-interacting entities. TRIM29 not only directly interacts with PERK but also orchestrates its SUMOylation, thereby preserving its stability and fostering PERK-mediated pathways. Intriguingly, the PERK inhibitor GSK2656157 was found to disrupt the TRIM29-pPERK nexus, mitigating PERK-involved ER stress, apoptosis, and TRIM29-induced PERK SUMOylation, while simultaneously curtailing CVB3 and EMCV replication in vitro. Such findings inspired in vivo therapeutic investigations, wherein GSK2656157 treatment substantially alleviated viral myocarditis. This amelioration was characterized by enhanced cardiac performance, fortified cardiac antiviral mechanisms, and a decline in immunosuppressive mMDSC to augment functions of antiviral CD8 T cells. Our findings about mMDSC align with prior report indicating that MDSC plays a role in promoting the development and progression of viral myocarditis[54]. Furthermore, we think that the true role of TRIM29-PERK axis is the influence on the virus, which means less cardiotropic viruses lead to less viral myocarditis disease. In essence, our study pioneers the illumination of TRIM29-driven PERK signaling pathways in viral myocarditis pathogenesis, underscoring the therapeutic potential of targeting the TRIM29-PERK axis to combat the disease.

Apoptosis is programmed cell death and is a double-edged sword for host and pathogen survival[55]. While it's established that CVB3 infection in the heart triggers cardiomyocyte apoptosis[56], there's been a void in our understanding of whether this apoptotic response is a universal reaction to all cardiotropic viruses and the intricate mechanisms that activate apoptosis during viral myocarditis. Our study sheds light on this by demonstrating that the cardiotropic viruses CVB3 and EMCV induce apoptosis in both human and mouse cardiomyocytes and that the PERK/ATF4/CHOP signaling pathway is required for cardiotropic virus-induced apoptosis. Although apoptosis can be initiated by the host to limit virus propagation, we showed that cardiotropic viruses induce cardiomyocyte apoptosis to cause the pathogenesis of viral myocarditis. SARS-CoV-2 infection can induce apoptosis in lung epithelial cells and lungs to cause pathogenesis both in vitro and in vivo[57–59], which is consistent with our findings. ROS also act as a double-edged sword for pathogen infection because some pathogens are effectively controlled by ROS, while other pathogens thrive in a cellular environment with abundant ROS[60]. In the context of our study, we found that cardiotropic virus infection triggers ROS, which further oxidizes TBK1 and inhibits TBK1-mediated type I IFN production to enhance viral replication in both human and mouse cardiomyocytes. Our data suggest that cardiotropic viruses could hijack PERK-mediated apoptosis and ROS responses to promote their replication in cardiomyocytes.

PERK is the second major branch of the ER stress response and is involved in the pathogenesis of cancer[20], diabetes[61] and neurodegenerative diseases[62]. However, there are few reports about the roles of PERK in cardiovascular diseases. While PERK has been shown to offer protective effects in pressure overload-induced heart failure and lung remodeling[63], our findings illuminates its pathogenic role in viral myocarditis, which aligns with its documented deleterious roles in cancer[20], diabetes[61] and neurodegenerative diseases[62]. The complex, condition-dependent role of ER stress in the development of multiple hepatic diseases[64] led us to suspect that different PERK signaling pathways are triggered in pressure overload-induced heart failure and viral myocarditis. TRIM29 is an E3 ubiquitin ligase, and TRIM29-mediated protein ubiquitination has been shown to play pivotal roles in antiviral innate immunity[51–53,65] and cancer development[50]. In our study, we demonstrated that TRIM29 could function as a SUMO E3 ligase to bind the SUMO-conjugating enzyme Ubc9 E2 to induce SUMOylation of PERK, which further maintains its stability. To our knowledge, this report presents important findings on the

SUMOylation of PERK. Given PERK's involvement in the onset of diseases like cancer, diabetes, and neurodegenerative disorders, it's tempting to postulate that the TRIM29-mediated SUMOylation of PERK might be a key regulator in their respective pathogenesis.

Given the diverse mechanisms of infection and pathogenesis employed by the myriad DNA and RNA viruses responsible for viral myocarditis, devising virus-specific therapies remains a formidable challenge. In this context, host-directed immunotherapy[66] emerges as a promising avenue for treating viral myocarditis. Our research indicates that TRIM29 is an essential modulator of antiviral innate immunity against a range of DNA and RNA viruses, specifically in dendritic cells and macrophages[51–53]. This positions TRIM29 as a viable target for host-directed immunotherapy of viral myocarditis. Our findings further elucidate that the PERK inhibitor, GSK2656157, can mitigate viral myocarditis by interfering with the TRIM29-PERK axis. This improves cardiac function, augments cardiac antiviral responses, attenuates inflammation, diminishes cardiac ER stress and apoptosis, and reduces immunosuppressive mMDSC to enhance functions of antiviral CD8 T cells in vivo. Given the importance of PERK in the pathogenesis of cancer, diabetes, and neurodegenerative diseases, the TRIM29-PERK axis may be therapeutically targeted to treat those diseases.

In summary, our research offers unprecedented insights into the role of TRIM29-mediated PERK signaling in the pathogenesis of viral myocarditis, underscoring the therapeutic efficacy of PERK inhibitors for its treatment. Our groundbreaking discovery suggests that targeting the TRIM29-PERK axis could serve as a potent therapeutic strategy not only for viral myocarditis but also for other infectious diseases linked to PERK.

## Methods
### Ethics statement
Animal work in this study was performed in accordance with the National Institute of Health Guide for the Care and Use of Laboratory Animals and the Animal Welfare Act. All protocols were ethically reviewed and approved by the Houston Methodist Institutional Animal Care and Use Committee (IACUC, #IS00007198). We purchased frozen heart tissue slides from normal human adults (Cat #: T1234133, BioChain) and frozen heart tissue slides from human patients with cardiomyopathy (Cat #: CS616191, ORIGENE) for immunohistochemistry (IHC) staining.

### Mice
*Trim29*[-/-] mice were obtained from the European Mouse Mutant Archive (EMMA)[51–53]. *Trim29*-targeted mice were crossed with FLP-deleted mice (B6 ROSA26Flpo; Stock No: 012930, The Jackson Laboratory) to produce *Trim29*[fl/fl] mice[65]. The *Trim29*[fl/fl] mice were further crossed with *αMyHC-Cre* (WTMyHC-Cre) transgenic mice (Stock No: 011038, The Jackson Laboratory) that express Cre recombinase under the transcriptional control of the cardiomyocyte-specific α-myosin heavy chain (αMyHC) promoter[67] to generate cardiomyocyte-specific Trim29 knockout mice, *Trim29*[fl/fl]; *αMyHC-Cre* (*Trim29*[MyHC-KO]). We used age matched male mice in all animal experiments. All animals were on the C57BL/6 genetic background and maintained in the specific pathogen-free facility at Houston Methodist Research Institute in Houston, Texas. Animals were housed under the following conditions: temperatures of 68–72 F, 30–70% humidity, 10–15 fresh air exchanges hourly, and a 12:12 h light:dark cycle (lights were on from 07:00–19:00). Mice were housed in sterile individually-ventilated cages (Techniplast S.p.A., Buguggiate, Italy) containing autoclaved Bed-o'Cobs 1/4" bedding (The Andersons, Inc.), a sterile cotton nesting square or sterile crinkle nesting material, and received approximately 75 air changes hourly. Mice were housed at a density of up to five mice.

### Reagents
The following antibodies were used for immunoblot analysis: anti-TRIM29 (1:1000; A301-210A; Bethyl), anti-TRIM29 (1:1000; sc-33151;

Santa Cruz), anti-ANP (1:1000; sc-515701; Santa Cruz), anti-FITC (1:2000; 71–1900; Thermo Fisher Scientific), antibody to phosphorylated PERK (1:1000; MA5-15033; Thermo Fisher Scientific), anti-PERK (1:1000; 3192 S; Cell Signaling Technology), anti-IRE1α (1:1000; 3194 S; Cell Signaling Technology), anti-ATF6 (1:1000; 65880 S; Cell Signaling Technology), anti-CHOP (1:1000; 2895 S; Cell Signaling Technology), anti-cleaved caspase-3 (1:1000; 9664 S; Cell Signaling Technology), anti-BCL2 (1:1000; 3498 S; Cell Signaling Technology), anti-BAX (1:1000; 2772 S; Cell Signaling Technology), anti-STING (1:1000; 13647 S; Cell Signaling Technology), anti-IRF3 (1:1000; 4302 S; Cell Signaling Technology), anti-TBK1 (1:1000; 3504 S; Cell Signaling Technology), anti-SUMO1 (1:1000; 4930 S; Cell Signaling Technology), anti-SUMO2/3 (1:1000; 4971 S; Cell Signaling Technology), antibody to phosphorylated IRF3 at Ser396 (1:1000; 4947 S; Cell Signaling Technology), antibody to phosphorylated TBK1 at Ser172 (1:1000; 5483 S; Cell Signaling Technology), antibody to phosphorylated IRE1α (1:1000; ab48187, Abcam), anti-His (1:5000; 652504; BioLegend), anti-Flag (1:5000; A8592; Sigma), anti-GAPDH (1:20,000; G9295; Sigma), anti-HA (1:5000; H6533; Sigma) and anti-Myc (1:5000; 16–213; Sigma). The anti-KDEL ER marker antibody (1:500; sc-58774, Santa Cruz) was used for immunohistochemistry (IHC) analysis. Anti-HA and anti-Myc agarose beads were purchased from Sigma. The biotin anti-Gr1 antibody (108404) and MojoSort Streptavidin Nanobeads (480016) were purchased from BioLegend. A mouse T cell isolation kit (130-095-130) was purchased from Miltenyi Biotec. Lentiviral vectors for shRNA were from Dharmacon Inc. (Horizon Discovery Group company): human TRIM29 (clone TRCN0000016352). The DCFDA/H2DCFDA-Cellular ROS Assay Kit (ab113851) was purchased from Abcam. The CellTiter-Glo® Luminescent Cell Viability Assay was from Promega. Ni²⁺-NTA agarose (R90101) was purchased from Thermo Fisher Scientific. A mouse creatine kinase ELISA kit (NBP2-75306) was purchased from Novus Biologics. The human IFN-β (41435), mouse IFN-β (42410), human IFN-α (41100) and mouse IFN-α (42120) ELISA kits were from PBL Interferon Source. The mouse TNF-α (DY410), IL-6 (DY406) and IL-1β (DY401) ELISA kits were from R&D Systems. The selective PERK inhibitor GSK2656157 (HY-13820) was purchased from MedChemExpress. Coxsackievirus B3 (CVB3, strain Nancy) was obtained from ATCC (ATCC® VR-30™). The encephalomyocarditis virus (EMCV) was a gift from Dr. Michael S. Diamond (Washington University in St. Louis). The ACK lysing buffer (A1049201) was purchased from Thermo Fisher Scientific.

## Cell culture
The AC16 human cardiomyocyte cell line (Cat. No.: SCC109) is derived from adult human ventricular cardiomyocytes and was purchased from EMD Millipore. Human cardiomyocytes and HEK293T cells (CRL-3216, ATCC) were cultured in Dulbecco's modified Eagle's medium (DMEM) supplemented with 10% heat-inactivated fetal bovine serum (FBS) and 1% penicillin-streptomycin (Invitrogen-Gibco)[68,69].

## Isolation of neonatal mouse cardiomyocytes
Neonatal mouse cardiomyocytes were prepared from 2-day-old Trim29[+/+] and Trim29[-/-] C57BL/6 mice using the Pierce primary cardiomyocytes isolation kit (#88281, Thermo Fisher Scientific)[70]. In brief, day 2 post-neonate mice were sacrificed, and the hearts were isolated. After mincing each heart into 1–3 mm³ pieces, the tissues were washed twice with HBSS (Hanks-based salt solution), resuspended, and incubated in working solution including primary cardiomyocyte isolation enzymes 1 and 2 (components of the kit) at 37 °C for 30 min. The tissues were washed with HBSS several times and then with DMEM. Cells were plated at a density of $1.25 \times 10^6$ per well in six-well plates for 2 h to separate cardiac fibroblasts (rapidly adhering) from cardiomyocytes (still floating at 2 h after plating). For cardiomyocyte culture, 24 h after plating the cells, fresh medium was added with cardiomyocyte differentiation supplement (another component of the kit). After

7 days of growth and differentiation with one medium change on day 4, cells were used as primary cardiomyocytes.

## Lentivirus transduction and virus infection
The pLKO.1 lentiviral vector carrying a scrambled shRNA or target gene sequences (Dharmacon Inc.) were cotransfected into HEK 293FT cells with the packaging plasmids psPAX2 (Addgene 14858) and pMD2. G (Addgene 12259) using Lipofectamine 3000 (Thermo Fisher Scientific) to produce lentivirus. AC16 human cardiomyocytes were infected by lentivirus[71]. After 24 h of culture, cells were selected by the addition of puromycin (2 ng/ml) to the medium. The knockdown efficiency was detected with immunoblot analysis[72]. Human cardiomyocytes after lentivirus transduction or mouse primary cardiomyocytes from Trim29[+/+] and Trim29[-/-] mice were infected by cardiotropic viruses CVB3 or EMCV at a multiplicity of infection (MOI) of 1[73]. The virus-induced ER stress, apoptosis and ROS responses in human or mouse cardiomyocytes were assessed by immunoblot, qRT-PCR and ROS measurement assays. The concentrations of IFN-α and IFN-β in culture supernatants were measured by ELISA[74].

## Immunohistochemical analysis
For immunohistochemistry (IHC) staining of human hearts, we purchased frozen heart tissue slides from normal human adults (Cat #: T1234133, BioChain) and frozen heart tissue slides from human patients with cardiomyopathy (Cat #: CS616191, ORIGENE). For IHC staining of human and mouse hearts, paraffin-embedded hearts were cut transversely into 5-µm sections. Following a 5-min high-pressure antigen retrieval process in citrate buffer with a pH of 6.0, the heart sections were blocked with 10% bovine serum albumin for 60 min and were subsequently incubated overnight at 4 °C with primary antibodies, including anti-KDEL ER marker rabbit antibody (BS-6940R, Thermo Fisher Scientific) and anti-TRIM29 rabbit antibody (sc-33151; Santa Cruz). Binding was visualized with the appropriate peroxidase-conjugated secondary antibodies (Horseradish Peroxidase AffiniPure Goat Anti-Rabbit IgG (H + L), 111-035-003, Jackson ImmunoResearch) for 30 min at 37 °C[74].

## Quantitative RT-PCR
RNA was isolated using the RNeasy Kit (Qiagen) according to the manufacturer's instructions. The isolated RNA was used to synthesize cDNA with the iScript cDNA Synthesis Kit (Bio-Rad)[51]. Quantitative RT-PCR (qRT-PCR) was performed on a CFX-96 real-time PCR detection system (Bio-Rad) with iTaq Universal SYBR Green Supermix (Bio-Rad)[75]. All qRT-PCR primers are listed in Supplementary Table 3.

## Cell viability assay
Cell viability was quantified with the CellTiter-Glo assay (Promega). Cells were lysed together with culture supernatant at a 1:1 ratio (volume) with CellTiter-Glo reagent and incubated at room temperature for 10 min, followed by measurement of the luminescence signal with a microplate reader at excitation/emission wavelengths of 485 nm/520 nm (BioTek Synergy H1, Winooski, VT).

## ROS measurement assay
Total cellular ROS measurement in human or mouse cardiomyocytes was analyzed using the DCFDA/H2DCFDA - Cellular ROS Assay Kit (ab113851, Abcam). Briefly, human or mouse cardiomyocytes were plated in 96-well plates and infected with CVB3 or EMCV for 6 h. After infection, cardiomyocytes were washed with PBS, stained with a DCFDA (10 µM) probe for 30 min, and quickly analyzed with a microplate reader with excitation/emission at 485 nm/535 nm (BioTek Synergy H1, Winooski, VT).

## Detection of oxidative modification of TBK1
AC16 human cardiomyocytes were first treated with an shRNA for TRIM29 knockdown expression (sh-T29) or a scrambled control

shRNA (sh-Ctrl). Next, shRNA-treated human cardiomyocytes or mouse cardiomyocytes isolated from *Trim29*[+/+] and *Trim29*[−/−] mice were infected without or with CVB3 and EMCV for 6 h at an MOI of 5. Human and mouse cardiomyocytes were then washed with ice-cold phosphate-buffered saline (PBS) twice and lysed with gentle cell lysis buffer (9803, Cell Signaling Technology) containing 5 mM 5-IAF, which labels free thiols with a fluorescein (FITC) tag[76]. The cell lysate was spun down at 16,000 g for 10 min, and the supernatant was then incubated at room temperature for 1 h in the dark. Lysates were subjected to immunoprecipitation for TBK1 using an anti-TBK1 antibody and probed for both TBK1 and FITC using immunoblotting.

## In vivo virus infection

For the in vivo CVB3-induced myocarditis study, age matched *Trim29*[+/+], *Trim29*[−/−], *Trim29*[fl/fl], WTMyHC-Cre, and *Trim29*[MyHC-KO] male mice (n = 10 per strain, 6 weeks old) were injected intraperitoneally with CVB3 (1 × 10^7 PFU/mouse)[74]. For therapeutic treatment of viral myocarditis using a PERK inhibitor, age matched *Trim29*[+/+] male mice (n = 10 per strain, 6 weeks old) were injected intraperitoneally with CVB3 (1 × 10^7 PFU/mouse), followed by intraperitoneal injection with the PERK inhibitor GSK2656157 (25 mg/kg per day) or dimethyl sulfoxide (DMSO) control for 5 days or until sample harvest. The survival of mice was monitored daily for 14 days after CVB3 infection. On days 2 and 4 after infection, mice were euthanized, and whole hearts were excised into PBS and homogenized to determine viral titers and concentrations of cytokines by ELISA. Additionally, the heart weight was determined by scale before and after CVB3 infection for 6 days.

## Virus titration

After CVB3 infection in mice, total heart tissues were removed and homogenized to prepare heart extracts in 1 ml of PBS (pH 7.4). The supernatants from the homogenized heart tissues were diluted and then used to infect confluent HeLa cells (ATCC® CCL-2™) cultured on 12-well plates[74]. At 1 h postinfection, the supernatant was removed, and 2% low melting-point agarose was overlaid. At 3 days postinfection, the overlay was removed, and the cells were fixed with methanol:acetic acid solution (3:1 methanol:acetic acid) for 20 min and stained with 0.2% crystal violet[69]. Plaques were counted, averaged, and multiplied by the dilution factor to determine viral titer[71].

## Echocardiography measurement

Transthoracic echocardiography was performed on day 0 before infection and day 4 after CVB3 inoculation. Mice were anesthetized by isoflurane inhalation. A comprehensive echocardiographic study was performed, including 2-dimensional imaging and M-mode imaging using the Vevo 2100 system (VisualSonics, Toronto, Canada)[74]. Lubricant was applied to the mice's eyes to prevent dryness. The imaging target area was prepared by applying a hair removal cream followed by thorough cleaning with a cotton pad. To ensure stability, the core temperature was maintained at 37 °C, heart rates were kept consistent between groups throughout the procedure. Imaging commenced with the transducer aligned along the long axis of the left ventricle (LV) for a two-dimensional B-Mode view. It was then rotated 90° clockwise for the LV short-axis view, ensuring clarity of the papillary muscles. The system was switched from B-Mode to M-Mode to capture the images. Post-imaging, excess gel was removed and mice were allowed to recover in their cages. Echo images were analyzed for ejection fraction (EF) and fractional shortening (FS) using the Vevo LAB 5.6.0 software.

## Histology

Hearts and pancreas were removed from mock- and CVB3-infected *Trim29*[+/+], *Trim29*[−/−], *Trim29*[fl/fl] and *Trim29*[MyHC-KO] male mice. These removed heart and pancreas tissues were washed using PBS before being fixed with 10% formaldehyde for 24 h at room temperature[51]. The tissues were embedded in paraffin and processed by standard techniques. Longitudinal 5-μm sections were stained with hematoxylin & eosin (H&E)[74].

## In vitro coimmunoprecipitation and immunoblot analysis

For the preparation of purified PERK and TRIM29, HEK293T cells were transfected with expression plasmids encoding full-length or truncated versions of HA- or Myc-tagged PERK or TRIM29. Lysates were prepared from the transfected cells, followed by incubation with anti-HA or anti-Myc beads. Proteins were eluted from the beads after beads were washed six times with PBS. For precipitation with anti-HA or anti-Myc beads, purified HA-tagged wild-type PERK or truncations of PERK were incubated for 2 h with purified Myc-tagged TRIM29 or purified HA-tagged TRIM29 or truncations of TRIM29 were incubated for 2 h with purified Myc-tagged PERK. Beads were added; after 2 h of incubation, the bound complexes were pelleted by centrifugation. Proteins and beads were analyzed by immunoblot analysis with anti-HA or anti-Myc Abs. For immunoprecipitation of endogenous proteins, whole-cell lysates of *Trim29*[fl/fl], WTMyHC-Cre, and *Trim29*[MyHC-KO] mouse cardiomyocytes left infected (Mock) or infected with CVB3 or EMCV were incubated with anti-PERK or immunoglobulin G antibodies. After 2 h of incubation, the protein A/G beads were added for another 3 h incubation, and the bound complexes were pelleted by centrifugation. Proteins and beads were analyzed by immunoblot analysis with anti-TRIM29 and anti-PERK antibodies. Human and mouse cardiomyocytes were left infected or infected with CVB3 or EMCV for the indicated times and were then lysed in 1% NP-40 lysis buffer (50 mM Tris-HCl, 1% NP-40, 0.1% SDS, 150 mM NaCl) supplemented with protease inhibitor (Thermo Fisher Scientific) followed by centrifugation. Supernatants were collected and incubated with SDS sample buffer by boiling of samples for 8 min followed by SDS-PAGE and immunoblot analysis[51].

## Protein SUMOylation assays

Cells cultured in 6-cm plates were transfected with the indicated plasmids. After transfection for 24 h, cells from each plate were collected and divided into two aliquots. One aliquot was lysed in lysis buffer and analyzed by immunoblot to examine the expression of transfected proteins. Another aliquot was lysed in buffer A (6 M guanidinium-HCl, 0.1 M $Na_2HPO_4/NaH_2PO_4$, 10 mM Tris-Cl pH 8.0, 5 mM imidazole, and 10 mM β-mercaptoethanol) and incubated with $Ni^{2+}$-NTA agarose beads (Thermo Fisher Scientific) for 4 h at room temperature or overnight at 4 °C. The beads were washed sequentially with buffers A and B (8 M urea, 0.1 M $Na_2HPO_4/NaH_2PO_4$, 10 mM Tris-HCl pH 8.0, 10 mM β-mercaptoethanol), and C (same as B except pH = 6.3). Beads with bound proteins were then boiled in SDS sample buffer, and the proteins were fractioned by SDS-PAGE and analyzed by immunoblot.

## MDSC isolation and T cell proliferation assays

Mouse CD3[+] T cells were enriched through negative selection using a mouse T cell isolation kit (Miltenyi Biotec) from the spleens of wild-type mice[77]. MDSC infiltrated in hearts from CVB3-infected *Trim29*[fl/fl] and *Trim29*[MyHC-KO] male mice were isolated from cellular suspensions of digested hearts followed by positive selection (MojoSort Streptavidin Nanobeads, BioLegend) using biotin anti-Gr1 antibody (108404, BioLegend). Murine heart MDSC (ratio 1:1/4) were cocultured with negatively selected CD3[+] T cells in a 96-well plate bound with anti-CD3 and anti-CD28 antibodies (1 μg/ml each, BioLegend) for 3 days. The CD8 T cell proliferation was assessed with carboxyfluorescein succinimidyl ester (CFSE) dye dilution detected by flow cytometry.

## Flow cytometry

Mouse spleens and hearts were collected from CVB3-infected *Trim29*[fl/fl], WTMyHC-Cre, and *Trim29*[MyHC-KO] male mice. Preparation of cells from the spleen was performed by mechanical disaggregation of the tissue

through a 100 μm strainer using a syringe plunger. The cell suspension was then moved to a collection tube, and the 70 μm cell strainer was washed twice with ice-cold PBS, followed by incubation in red blood cell (RBC) lysis buffer with ACK lysis buffer (A1049201, Thermo Fisher Scientific) to remove the erythrocytes from the cell suspension. For each spleen, 1 ml of room temperature ACK lysis buffer was added, and the tube was shaken manually for 2 min before washing with ice-cold PBS[71]. The cell suspensions were then used for flow cytometry.

Hearts were perfused with PBS and then minced using a razor blade. The minced hearts were then incubated with 2 mL of tissue digestion enzyme solution with 3000 U/ml collagenase II and 90 U/ml DNase I (Sigma) for 30 min at 37 °C in 35 mm dishes. After incubation with digestion enzymes, tissues were dissociated using a gentleMACS Dissociator (Miltenyi). Cells were washed and filtered through 70 μm cell strainers. The filtered cells were suspended in 3 ml of 40% Percoll and centrifuged at $600 \times g$ (acceleration: 6, deceleration: 1) for 10 min at room temperature to isolate heart infiltrated mononuclear cells[72]. The cell suspensions were then used for flow cytometry.

For intracellular cytokine staining, cells were stimulated in vitro for 4 h with phorbol 12-myristate 13-acetate (50 ng/ml) and ionomycin (550 ng/ml; Sigma-Aldrich) in the presence of GolgiStop (BD Biosciences) before staining. Viability was determined by LIVE/DEAD staining using a Zombie Aqua fixable viability kit (423102, BioLegend). For intracellular cytokine staining, cells were resuspended in fixation and permeabilization solution (00-5523-00, eBioscience). Cells were blocked with anti-CD16/CD32 antibody (101320, BioLegend) and stained with fluorochrome-conjugated monoclonal antibodies. The following antibodies were used to analyze the composition of mMDSC, macrophages, neutrophils, B cells, CD4$^+$ and CD8$^+$ T cells: APC/Cyanine7 anti-mouse CD45 antibody (103116, BioLegend), Brilliant Violet 785 anti-mouse/human CD11b antibody (101243, BioLegend), APC anti-mouse Gr1 antibody (108412, BioLegend), PE anti-mouse Ly6C antibody (128008, BioLegend), FITC anti-mouse Ly6G antibody (127606, BioLegend), Brilliant Violet 421 anti-mouse CD64 antibody (164407, BioLegend), FITC anti-mouse CD3 antibody (100204, BioLegend), PE anti-mouse CD19 antibody (152408, BioLegend), PE/Cyanine7 anti-mouse CD4 antibody (100528, Biolegend), PerCP/Cyanine5.5 anti-mouse CD8a antibody (100734, BioLegend) and APC anti-mouse IFN-γ antibody (505810, BioLegend). Flow cytometry data were acquired on an LSR-II flow cytometer (Beckton Dickinson) and analyzed using FlowJo v10 software (Tree Start)[71].

## Quantification and statistical analysis

No statistical methods were used to predetermine sample sizes, but our sample sizes are similar to those reported in previous publications[51,53,71]. Data are shown as the mean ± SD unless stated otherwise. Statistical analyses were performed using GraphPad Prism9 software (GraphPad Software, Inc.). Statistical significance was calculated using Student's two-tailed unpaired $t$ test. The log-rank (Mantel-Cox) test was used for survival comparisons. Differences were statistically significant when P values were less than 0.05.

## Reporting summary

Further information on research design is available in the Nature Portfolio Reporting Summary linked to this article.

# Data availability

All data generated or analyzed during the study are included in this article and its Supplementary Information. Uncropped and unprocessed scans of blots have been provided as in the Source data file. Source data are provided with this paper.

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

## Acknowledgements

We thank the Wellcome Trust Sanger Institute Mouse Genetics Project (Sanger MGP) and its funders for the mutant mouse line Trim29 and the European Mouse Mutant Archive (http://www.emmanet.org) partner from which the mouse line was received. Funding and associated primary phenotypic information are provided at the Sanger website (http://www.sanger.ac.uk/mouseportal). We thank Dr. Michael S. Diamond (Washington University in St. Louis) for EMCV virus, Dr. David Ribet (Université de Rouen, France) for plasmids His-SUMO1, His-SUMO2, His-SUMO3 and pCDNA3-FLAG-Ubc9. We thank the animal facility and the flow cytometry core at Houston Methodist for excellent services, and we thank Laurie J. Minze for operational support. We thank Dr. Zhiqiang Zhang, Dr. Xian Chang Li, Dr. John P. Cooke, and Dr. Dorothy E. Lewis at Houston Methodist for providing advice of this project and critically editing this manuscript. This work was supported by the American Heart Association Career Development Award 20CDA35260116 and Transformational Project Award 23TPA1055437 (https://doi.org/10.58275/AHA.23TPA1055437.pc.gr.172259) (J.X.).

## Author contributions

J.W., W.L., J.Z., Y.D. and J.X. designed and performed the experiments; A.Z., M.F., G.S. and S.C. helped with some of the experiments; J.X. conceived the idea and supervised the project; J.X. wrote the manuscript.

## Competing interests

The authors declare no competing interests.
