## [Peer Review File · Nature Communications]

Loss of TRIM29 mitigates viral myocarditis by attenuating PERK-driven ER stress response in male miceReviewer #2 (Remarks to the Author):

In the manuscript by Wang et al., the authors sought to better understand the role of the TRIM29-PERK axis in RNA virus infection and virus pathogenesis. The manuscript and experiments are well presented and designed. They chose the model of CVB3 infection and myocarditis to study and asked whether the TRIM29-PERK axis was engaged during acute infection, what the effects on virus replication were and did this translate to cardiac pathology? Using systemic knock-out mice and cardiac specific ko mice they demonstrated that there was a direct influence on the replication of the virus in the heart and subsequent myocarditis. Their results describe the virus's ability to induce TRIM29 function concomitant with acute infection, and that removal of TRIM29 by KO, systemic or cardiac specific, protected the mouse from virus replication and cardiac specific damage. Intriguingly, TRIM29 promotes ER stress and host regulation of the immune response, which seems to be at odds and a unique niche that the virus has found. Increased regulation reduces the adaptive response and allows the virus to replicate and cause more damage. The paper examines an inhibitor of TRIM29-PERK and the authors demonstrate protection from infection and disease as a clear result from reduced immunosuppression and thereby increased adaptive responses. These studies are well justified and controlled. The results are significant and likely speak to a wider range of viruses and their ability to cause pathogenesis.

Important questions and concerns:

- 1) CVB3 infects the mice in other organs besides the heart including the pancreas, spleen, liver, etc,. Often lethal infection is due to destruction of the pancreas. What were the virus titers in these organs and how did the KO and fl/fl ko alter these? Did they have an effect on other disease pathology?
- 2) The authors believe that the TRIM29-PERK axis is essential for virus replication and its true role is the influence on the virus. The role on myocarditis follows with replication. Less virus, less disease. Yet in many places including the title, the authors jump from promoting the axis means more disease or inhibiting means less. I think this should be made clearer unless the authors have data that suggests less or more disease without the presence of the virus.
- 3) The use of the term "MHC-KO" for the floxed mouse is confusing as MHC can refer to other important immune molecules. I suggest changing this to MyHC or something similar.
- 4) The term MDSC is not well defined in the paper.
- 5) What do the authors consider to be the mechanism behind increased immunosuppression by the TRIM29-PERK axis and is this path supposed to protect us from self-reactive responses to normal cell turnover versus pathogen driven apoptosis? How are these paths different? This could be added to the discussion.

Reviewer #3 (Remarks to the Author):

In this manuscript the Authors define the role of a TRIM29-PERK axis and ER stress in the pathogenetic mechanisms leading to viral myocarditis. In infected cardiomyocytes, they show that TRIM29 interacts with PERK to promote ER stress and enhance viral replication. TRIM29 promotes SUMOylation of PERK to maintain its stability in infected cells. By interfering with TRIM29-PERK binding in vivo, the Authors observed improvement of cardiac function and increased antiviral innate immunity.

The work has high significance in terms of basic knowledge and also of translational data towards possible therapeutic advances. I think, however, that some claims should be more strongly supported, as for the comments here below.

Major comments

1. The proper control of Figure 5a should be non-infected cells where, potentially, both proteins could be present, rather than TRIM29 cKO that genetically cannot express the protein.
2. GSK2656157 binds to PERK active site; how this relates to impaired TRIM29 binding? What

should be investigated is if phospho-PERK form is bound and stabilized by TRIM29. In the proposed model PERK auto-phosphorylation follows TRIM29 binding but this piece of data is not proven and should be studied in order to explain the effect of the inhibitor on the interaction.

3. As few SUMO E3 ligases have been assessed to date and they employ either Zn-finger-like domains or SUMO Interacting Motifs (SIMs) to catalyze the transfer, it is quite unexpected that the binding with Ubc9 is not mediated by the N-terminal part of TRIM29 where the B-box resides. The complementary IP with a TRIM29 C-terminal portion should be integrated in Figure 6a.

4. Along the same line as above, the authors should show if the sequence around the Lysine residues modified by SUMO are consensus sites for SUMOylation (often independent of E3 ligases) and if SIMs are present in TRIM29. It is also possible that TRIM29 does not 'catalyze' but 'promotes' PERK SUMOylation. This should be changed accordingly.

5. As SUMO1 does not form poly-SUMO chains and more than 2 SUMO-PERK bands are observed, mono-SUMO1 on these two Lys is not enough to account what observed in gels. It should be checked whether on the blot also some SUMO2/3 moieties are present in mixed chains with SUMO1.

6. One would expect to see unstable/less abundant PERK in the input of K641R/K672R mutants, why this is not the case?

Minor comments

A. The OmpH domain has been so far detected in bacteria and other unicellular organisms, can the author comment on its presence in TRIM29?

B. Check the MW in the shown blots, e.g. MycTRIM29 in Figure 5e is shown at 2 different MW in IP and input.

Reviewer #4 (Remarks to the Author):

Wang et al. reported that TRIM29 plays a crucial role in the pathogenesis of viral myocarditis, a significant cause of sudden cardiac death in young adults. The authors found that TRIM29 expression is highly increased in virus-infected cardiomyocytes, leading to ER stress, apoptosis, and downregulation of type 1 interferon. Furthermore, they claimed that the absence of TRIM29 expression specifically in cardiomyocytes improved CVB3-induced myocarditis in mice by enhancing the activity of effector T cells and suppressing immunoregulatory cells such as MDSC and Treg cells. The authors discovered that TRIM29 is essential for stabilizing PERK, downstream of TRIM29, through SUMOylation contributing to the pathogenesis during viral infection, providing a potential molecular mechanism. Thus, TRIM29 represents a potential therapeutic target for protecting the heart from viral myocarditis. The authors demonstrated the efficacy of GSK2656157, a TRIM29 inhibitor, in treating CVB3-induced myocarditis in vivo. The methods and data presented by the authors provide convincing support for their conclusion. However, several concerns should be addressed:

1. The authors primarily used Trim29 mutant mice for in vivo and in vitro experiments but presented the data with AC16 cardiomyocyte cell lines in Fig. 1 and 2. To improve clarity, it is recommended to switch the main figures and Extended Data figures, as similar findings were observed with primary cardiomyocytes isolated from mice in Extended Data Fig. 1 and 2.
2. The IHC stain for TRIM29 in the mouse heart in Extended Data Fig. 1a is convincing but there is no TRIM29 staining in the human heart in Fig. 1a. To strengthen the findings, the authors should present TRIM29 expression in the heart section from a cardiomyopathy patient.
3. It is worth noting that mice with cre alone may exhibit a phenotype different from that of WT mice, while mice with flox alone do not. Especially, myosin heavy chain (MHC)-cre mice were known to develop arrhythmia, cardiomyopathy, and cardiac fibrosis (PMID 30917606 and 36834504). To address this, have the authors used MHC-cre alone mice as a control in Fig. 3-6?

4. In Fig. 3 and 7, some groups showed 20% of survival along with 30% EF. However, the severity of myocarditis appears to be mild to moderate in these groups on H&E staining. It does not seem cardiac inflammation is a principal cause of death in these mice. The authors should consider further investigating ER stress, apoptosis, or other pathologies in cardiomyocytes in their in vivo models when Trim29 is deleted or inhibited. Additionally, in their in vitro studies, most of the data support that Trim29 is important for the pathogenesis of cardiomyocytes such as ER stress and apoptosis during viral infection, but they are not included in the in vivo studies.
5. In Fig. 4a-d, the authors claimed CD11b+Gr1+ cells as MDSC but it can be also a phenotype of neutrophils or monocytes, which are pro-inflammatory cells in the inflamed heart. The authors should confirm this by adding more markers for MDSC or testing gene signatures in sorted MDSC.
6. In Fig. 4e, FoxP3 staining in the heart of Trim29^{fl/fl} mice is not clear, while the Trim29MHC-KO mouse heart in the same figure or the spleen in Fig 4g exhibits quality staining. The authors should present a representative figure with better FoxP3 staining and consider adding CD25 as an additional marker for Treg cells.
7. In Fig. 4e-p, graphs show the proportion of FoxP3- or IFN-g-positive cells among CD4+ or CD8+ T cells. However, to better understand the immune profile in the heart with myocarditis, they should show the absolute number of major immune cell populations in the heart such as neutrophils, monocytes, macrophages, NK cells, B cells, CD4+ T cells, and CD8+ T cells.
8. In Fig. 4, the authors claimed that Trim29MHC-KO mice showed a reduced number of suppressor cells and increased activation of effector T cells in the heart compared to Trim29^{fl/fl} controls. Also, they found the same trend in the spleen, but this is cardiomyocyte-specific TRIM29 KO mice. The authors should provide an explanation as to know why spleen immune cells react to CVB3 infection in Trim29MHC-KO mice similarly to heart immune cells.
9. In Fig. 4i-p, the authors showed increased IFN-g production in T cells in Trim29MHC-KO mice compared to Trim29^{fl/fl} mice, however, the extent of inflammation shown in Fig. 3 by H&E staining and cytokine levels was significantly lower in Trim29MHC-KO. This is uncommon that T cells are highly activated during mild inflammation unless T cell-specific mutant mice are used for the study. The authors should address this discrepancy.
10. The authors showed only a gating strategy for the spleen but not for the heart in Extended Data Fig. 5. They should present a gating strategy for the heart.
11. This manuscript is readable, but it is strongly recommended to have it proofread by a native English speaker to improve clarity.

Point-by-point reply

We extend our gratitude to the reviewers for their insightful suggestions and constructive feedback. In response, we have conducted supplementary experiments, incorporated new data, and made corresponding revisions to the manuscript. We are confident that these amendments have not only addressed all raised concerns but have also substantially enhanced the manuscript's overall quality. For your convenience, all modifications in the revised version are highlighted in yellow. Below, we provide a detailed, point-by-point account of the revisions made to the manuscript.

REVIEWER COMMENTS

Reviewer #2 (Remarks to the Author):

In the manuscript by Wang et al., the authors sought to better understand the role of the TRIM29-PERK axis in RNA virus infection and virus pathogenesis. The manuscript and experiments are well presented and designed. They chose the model of CVB3 infection and myocarditis to study and asked whether the TRIM29-PERK axis was engaged during acute infection, what the effects on virus replication were and did this translate to cardiac pathology? Using systemic knock-out mice and cardiac specific ko mice they demonstrated that there was a direct influence on the replication of the virus in the heart and subsequent myocarditis. Their results describe the virus's ability to induce TRIM29 function concomitant with acute infection, and that removal of TRIM29 by KO, systemic or cardiac specific, protected the mouse from virus replication and cardiac specific damage. Intriguingly, TRIM29 promotes ER stress and host regulation of the immune response, which seems to be at odds and a unique niche that the virus has found. Increased regulation reduces the adaptive response and allows the virus to replicate and cause more damage. The paper examines an inhibitor of TRIM29-PERK and the authors demonstrate protection from infection and disease as a clear result from reduced immunosuppression and thereby increased adaptive responses. These studies are well justified and controlled. The results are significant and likely speak to a wider range of viruses and their ability to cause pathogenesis.

Reply: Thank you so much for your very positive comments regarding our work!

Important questions and concerns:

1) CVB3 infects the mice in other organs besides the heart including the pancreas, spleen, liver, etc.,. Often lethal infection is due to destruction of the pancreas. What were the virus titers in these organs and how did the KO and fl/fl ko alter these? Did they have an effect on other disease pathology?

Reply: Excellent points. We have provided new data to show that CVB3 viral loads were significantly reduced in heart, pancreas and spleen from Trim29^{-/-} or Trim29^{MyHC-KO} mice compared with those from Trim29^{+/+} or Trim29^{fl/fl} mice on day 2 after CVB3 infection in Supplementary Fig. 3h and Fig. 3h, respectively. Additionally, pancreas histopathology revealed that Trim29^{-/-} mice had significantly reduced inflammation and infiltration of inflammatory cells compared with Trim29^{+/+} mice following CVB3 infection in Supplementary Fig. 3b.

2) The authors believe that the TRIM29-PERK axis is essential for virus replication and its true role is the influence on the virus. The role on myocarditis follows with replication. Less virus, less disease. Yet in

many places including the title, the authors jump from promoting the axis means more disease or inhibiting means less. I think this should be made clearer unless the authors have data that suggests less or more disease without the presence of the virus.

Reply: Thank you for your thoughtful comments and suggestions. We totally agree with your comments. We also believe that the true role of TRIM29-PERK axis is the influence on the virus, less cardiotropic viruses lead to less viral myocarditis disease. We have revised them in the title and texts to make it clearer.

3) The use of the term "MHC-KO" for the floxed mouse is confusing as MHC can refer to other important immune molecules. I suggest changing this to MyHC or something similar.

Reply: Great suggestion. We have changed the term "MHC-KO" to "MyHC-KO" as suggested in the revised Figures and Texts.

4) The term MDSC is not well defined in the paper.

Reply: Excellent suggestion. MDSC are heterogeneous population of immature myeloid cells that include monocytic (mMDSC) and granulocytic (gMDSC) subsets. We have provided new data to demonstrate that these MDSC were expressed CD11b⁺Gr1⁺Ly6C⁺Ly6G⁻ in Fig 4e and 4f, suggesting they were mMDSC, but not gMDSC. We have renamed the term "MDSC" into "mMDSC" in the revised Figures and Texts.

5) What do the authors consider to be the mechanism behind increased immunosuppression by the TRIM29-PERK axis and is this path supposed to protect us from self-reactive responses to normal cell turnover versus pathogen driven apoptosis? How are these paths different? This could be added to the discussion.

Reply: We thank the reviewer for your excellent comments.

It's reported that PERK-active MDSC governs immunosuppression to promote tumor development by inhibiting STING signaling mediated antitumor immunity (PMID: 32294407). Additionally, MDSC is reported to promote the development and pathogenesis of viral myocarditis by restraining natural killer cell activity (PMID: 34065891). In this study, we showed that TRIM29-PERK axis increased mMDSC and promoted their immunosuppression functions to inhibit proliferation and function of antiviral CD8 T cells. We believe the TRIM29-PERK axis in cardiomyocytes promotes cardiotropic virus driven apoptosis and increases mMDSC-mediated immunosuppression leading to the pathogenesis of viral myocarditis, which has been added in the Discussion Section.

For the regulatory T cells (Treg) data, there is no direct evidence to show that PERK promotes Treg and their immunosuppression right now. We think the increase Treg may be the indirect outcome of high mMDSC. Given that the double-edged sword roles of immunosuppressive Treg in immune homeostasis, the increased Treg could possibly promote viral myocarditis by suppressing antiviral immune response or reduce viral myocarditis by protecting cells from self-reactive responses to normal cell turnover and resolving the virus induces inflammation. Right now, we did not provide direct evidence to show that the increased Treg could directly promote viral myocarditis through their immunosuppressive function. To avoid controversy, we decided to remove the increased Treg data from the revised manuscript.

Reviewer #3 (Remarks to the Author):

In this manuscript the Authors define the role of a TRIM29-PERK axis and ER stress in the pathogenetic mechanisms leading to viral myocarditis. In infected cardiomyocytes, they show that TRIM29 interacts with PERK to promote ER stress and enhance viral replication. TRIM29 promotes SUMOylation of PERK to maintain its stability in infected cells. By interfering with TRIM29-PERK binding in vivo, the Authors observed improvement of cardiac function and increased antiviral innate immunity.

The work has high significance in terms of basic knowledge and also of translational data towards possible therapeutic advances. I think, however, that some claims should be more strongly supported, as for the comments here below.

Reply: We greatly appreciate the reviewer's positive remarks regarding our work.

Major comments

1. The proper control of Figure 5a should be non-infected cells where, potentially, both proteins could be present, rather than TRIM29 cKO that genetically cannot express the protein.

Reply: Excellent suggestions. We have provided new coimmunoprecipitation data to show that the anti-PERK antibody, but not control IgG, precipitated endogenous TRIM29 in mouse cardiomyocytes from *Trim29^{fl/fl}* mice and *WTMyHC-Cre* mice after infection with CVB3 or EMCV, but not non-infected (Mock) cardiomyocytes in Fig. 5a.

2. GSK2656157 binds to PERK active site; how this relates to impaired TRIM29 binding? What should be investigated is if phospho-PERK form is bound and stabilized by TRIM29. In the proposed model PERK auto-phosphorylation follows TRIM29 binding but this piece of data is not proven and should be studied in order to explain the effect of the inhibitor on the interaction.

Reply: Excellent comments. We compared the interaction between TRIM29 and PERK or phosphorylated PERK (pPERK) in CVB3 infected WT cardiomyocytes treated with DMSO or PERK inhibitor GSK2656157 by coimmunoprecipitation assay using anti-PERK or anti-pPERK antibodies. We found there was still weak interaction of TRIM29 and PERK after GSK2656157 treatment (Below figure). However, PERK inhibitor GSK2656157 disrupted the interaction of TRIM29 and pPERK in CVB3 infected WT cardiomyocytes, which was provided as new data in Fig. 5g. GSK2656157 is a selective and ATP-competitive inhibitor of PERK. Because phosphorylation of PERK requires ATP. The interaction of pPERK and TRIM29 also requires ATP. We speculate that GSK2656157 treatment disrupted the interaction of TRIM29 and pPERK by competing with ATP. In addition, we have provided new data to show that overexpression of TRIM29 promoted the autophosphorylation and stability of PERK compared with vector control in Fig. 5f.

3. As few SUMO E3 ligases have been assessed to date and they employ either Zn-finger-like domains or SUMO Interacting Motifs (SIMs) to catalyze the transfer, it is quite unexpected that the binding with Ubc9 is not mediated by the N-terminal part of TRIM29 where the B-box resides. The complementary IP

with a TRIM29 C-terminal portion should be integrated in Figure 6a.

Reply: This is an excellent suggestion. We previously made a mistake to label “ΔN” as “N”. We compared together the interaction between Ubc9 and TRIM29 Full, mutant N and mutant C by coimmunoprecipitation assay. We demonstrated that both TRIM29 Full and mutant N, but not mutant C, interacted with Ubc9 in Fig. 6a. As expected, there is B-box domain that typically mediates binding of Ubc9 in TRIM29 mutant N.

4. Along the same line as above, the authors should show if the sequence around the Lysine residues modified by SUMO are consensus sites for SUMOylation (often independent of E3 ligases) and if SIMs are present in TRIM29. It is also possible that TRIM29 does not ‘catalyze’ but ‘promotes’ PERK SUMOylation. This should be changed accordingly.

Reply: Great suggestions. We found that there were two SUMO Interacting Motifs (SIMs), AA 140-143 and AA 282-288 in TRIM29 mutant N in Supplementary Table 2. As suggested, we have corrected the description that TRIM29 promotes PERK SUMOylation in Texts.

5. As SUMO1 does not form poly-SUMO chains and more than 2 SUMO-PERK bands are observed, mono-SUMO1 on these two Lys is not enough to account what observed in gels. It should be checked whether on the blot also some SUMO2/3 moieties are present in mixed chains with SUMO1.

Reply: We thank the reviewer for your excellent suggestion and have provided new evidence to show there were some SUMO2/3 moieties present in mixed chains with SUMO1 in Fig. 6d.

6. One would expect to see unstable/less abundant PERK in the input of K641R/K672R mutants, why this is not the case?

Reply: Great comments. Yes, we found there were unstable/less abundant PERK in the cell lysates from K641R, K672R or K641/K672R mutants compared with WT (See below figure). However, to avoid the possibility that the reduced PERK SUMOylation for K641R, K672R or K641/K672R mutants is caused by less input of K641R, K672R or K641/K672R mutants, we increased the input protein amounts of K641R, K672R or K641/K672R mutants to make the input comparable among WT PERK and those mutants.

Minor comments

A. The OmpH domain has been so far detected in bacteria and other unicellular organisms, can the author comment on its presence in TRIM29?

Reply: Great suggestion. We have previously shown that TRIM29 binds to NEMO via the

OmpH-OmpH domains to control antiviral and antibacterial innate immunity in alveolar macrophages (Xing et al. Nat Immunol. 2016;17(12):1373-1380). It's also reported recently that duck TRIM29 negatively regulates type I interferon production by targeting MAVS (PMID: 36685538). As reviewer mentioned, the OmpH domain has also been detected in bacteria and other unicellular organisms, suggesting that TRIM29 acquires this OmpH domain during its evolutionary history among bacteria and other unicellular organisms, bird and mammals including mouse and human. So, we speculate the critical role of TRIM29 in antiviral immunity in this study and previous studies may be well conserved in prokaryotes and eukaryotes.

B. Check the MW in the shown blots, e.g. MycTRIM29 in Figure 5e is shown at 2 different MW in IP and input.

Reply: We have corrected the wrong MW labeling in Fig. 5e.

Reviewer #4 (Remarks to the Author):

Wang et al. reported that TRIM29 plays a crucial role in the pathogenesis of viral myocarditis, a significant cause of sudden cardiac death in young adults. The authors found that TRIM29 expression is highly increased in virus-infected cardiomyocytes, leading to ER stress, apoptosis, and downregulation of type 1 interferon. Furthermore, they claimed that the absence of TRIM29 expression specifically in cardiomyocytes improved CVB3-induced myocarditis in mice by enhancing the activity of effector T cells and suppressing immunoregulatory cells such as MDSC and Treg cells. The authors discovered that TRIM29 is essential for stabilizing PERK, downstream of TRIM29, through SUMOylation contributing to the pathogenesis during viral infection, providing a potential molecular mechanism. Thus, TRIM29 represents a potential therapeutic target for protecting the heart from viral myocarditis. The authors demonstrated the efficacy of GSK2656157, a TRIM29 inhibitor, in treating CVB3-induced myocarditis in vivo. The methods and data presented by the authors provide convincing support for their conclusion. However, several concerns should be addressed:

Reply: We are grateful to the reviewer for the positive acknowledgement of our work.

1. The authors primarily used Trim29 mutant mice for in vivo and in vitro experiments but presented the data with AC16 cardiomyocyte cell lines in Fig. 1 and 2. To improve clarity, it is recommended to switch the main figures and Extended Data figures, as similar findings were observed with primary cardiomyocytes isolated from mice in Extended Data Fig. 1 and 2.

Reply: We thank the reviewer for your excellent suggestion, and we have switched the main figures Fig. 1 and Fig. 2 with Extended Data (now Supplementary) Figs. 1 and 2.

2. The IHC stain for TRIM29 in the mouse heart in Extended Data Fig. 1a is convincing but there is no TRIM29 staining in the human heart in Fig. 1a. To strengthen the findings, the authors should present TRIM29 expression in the heart section from a cardiomyopathy patient.

Reply: Great suggestion. We have now provided new IHC data to show that TRIM29 was highly induced in the hearts of human cardiomyopathy patients compared with healthy normal controls in Supplementary Fig. 1a.

3. It is worth noting that mice with cre alone may exhibit a phenotype different from that of WT mice, while mice with flox alone do not. Especially, myosin heavy chain (MHC)-cre mice were known to develop arrhythmia, cardiomyopathy, and cardiac fibrosis (PMID 30917606 and 36834504). To address this, have the authors used MHC-cre alone mice as a control in Fig. 3-6?

Reply: Excellent comments. We have provided new data to show that the WTMyHC-Cre mice exhibited a similar phenotype to those of WT *Trim29^{fl/fl}* mice in Supplementary Fig. 5, Supplementary Fig. 6, Fig. 5a and Fig. 6d.

4. In Fig. 3 and 7, some groups showed 20% of survival along with 30% EF. However, the severity of myocarditis appears to be mild to moderate in these groups on H&E staining. It does not seem cardiac inflammation is a principal cause of death in these mice. The authors should consider further investigating ER stress, apoptosis, or other pathologies in cardiomyocytes in their in vivo models when *Trim29* is deleted or inhibited. Additionally, in their in vitro studies, most of the data support that *Trim29* is important for the pathogenesis of cardiomyocytes such as ER stress and apoptosis during viral infection, but they are not included in the in vivo studies.

Reply: Excellent points. We have provided new evidence to show that there were less PERK-mediated ER stress and apoptosis in cardiomyocytes isolated from *Trim29^{MyHC-KO}* mice compared with wild-type *Trim29^{fl/fl}* and WTMyHC-Cre mice infected with CVB3 for 2 days in Supplementary Fig. 5h. Additionally, we showed that PERK-mediated ER stress and apoptosis were significantly reduced in cardiomyocytes isolated from CVB3 infected wild-type mice treated with PERK inhibitor GSK2656157 compared with mice with DMSO treatment Supplementary Fig. 9i.

5. In Fig. 4a-d, the authors claimed CD11b+Gr1+ cells as MDSC but it can be also a phenotype of neutrophils or monocytes, which are pro-inflammatory cells in the inflamed heart. The authors should confirm this by adding more markers for MDSC or testing gene signatures in sorted MDSC.

Reply: This is an excellent suggestion. MDSC are heterogeneous population of immature myeloid cells that include monocytic (mMDSC) and granulocytic (gMDSC) subsets. We have provided new data to demonstrate that these MDSC were expressed CD11b⁺Gr1⁺Ly6C⁺Ly6G⁻ in Fig 4e and 4f, suggesting they were mMDSC, but not gMDSC. We have renamed the term "MDSC" into "mMDSC" in the revised Figures and Texts.

6. In Fig. 4e, FoxP3 staining in the heart of *Trim29^{fl/fl}* mice is not clear, while the *Trim29^{MHC-KO}* mouse heart in the same figure or the spleen in Fig 4g exhibits quality staining. The authors should present a representative figure with better FoxP3 staining and consider adding CD25 as an additional marker for Treg cells.

Reply: Great suggestion. As we reply to Reviewer #2, for the regulatory T cells (Treg) data, there is no direct evidence to show that PERK promotes Treg and their immunosuppression right now. We think the increase Treg may be the indirect outcome of high mMDSC. Given that the double-edged sword roles of immunosuppressive Treg in immune homeostasis, the increased Treg could possibly promote viral myocarditis by suppressing antiviral immune response or reduce viral myocarditis by protecting cells from self-reactive responses to normal cell turnover and resolving the virus induces inflammation. Right now, we did not provide direct evidence to show that the

increased Treg could directly promote viral myocarditis through their immunosuppressive function. To avoid controversy, we decided to remove the increased Treg data from the revised manuscript.

7. In Fig. 4e-p, graphs show the proportion of FoxP3- or IFN-g-positive cells among CD4+ or CD8+ T cells. However, to better understand the immune profile in the heart with myocarditis, they should show the absolute number of major immune cell populations in the heart such as neutrophils, monocytes, macrophages, NK cells, B cells, CD4+ T cells, and CD8+ T cells.

Reply: Excellent suggestions. We have provided new data to show the absolute number of major immune cell populations including mMDSC, macrophages, neutrophils, B cells and T cells in Supplementary Fig. 6c.

8. In Fig. 4, the authors claimed that Trim29MHC-KO mice showed a reduced number of suppressor cells and increased activation of effector T cells in the heart compared to Trim29fl/fl controls. Also, they found the same trend in the spleen, but this is cardiomyocyte-specific TRIM29 KO mice. The authors should provide an explanation as to know why spleen immune cells react to CVB3 infection in Trim29MHC-KO mice similarly to heart immune cells.

Reply: This is a great point. Because the mice were infected by intraperitoneal injection, the spleen was one of major organs encountering the CVB3 virus. Additionally, there is blood circulation between spleen and heart. As shown in Fig. 3h and Supplementary Fig. 3h, CVB3 could also replicate in spleen of mice after CVB3 infection. That is why we also detect the immune cells in spleen of CVB3 infected mice. We also found similar trend in the spleen to that in the infiltrated immune cells in heart.

9. In Fig. 4i-p, the authors showed increased IFN-g production in T cells in Trim29MHC-KO mice compared to Trim29fl/fl mice, however, the extent of inflammation shown in Fig. 3 by H&E staining and cytokine levels was significantly lower in Trim29MHC-KO. This is uncommon that T cells are highly activated during mild inflammation unless T cell-specific mutant mice are used for the study. The authors should address this discrepancy.

Reply: We thank the reviewer for your excellent comments and suggestions. We detected the IFN- γ production in T cells *in vitro* using phorbol 12-myristate 13-acetate and ionomycin *in vitro* stimulation, which could not reflect the real IFN- γ production of T cells *in vivo*. That is why there is some discrepancy in the extent of inflammation shown in Trim29^{MyHC-KO} mice. The reason why we want to detect IFN- γ production in T cells *in vitro* is to further demonstrate the immunosuppressive mMDSC could inhibit the proliferation and function of T cells. Because there are many subpopulations of CD4 T cells, the role of IFN- γ production in CD4 T cells in controlling virus infection is different. Therefore, we removed the *in vitro* IFN- γ production in CD4 T cells from the revised manuscript. However, IFN- γ production in CD8 T cells is required to control virus infection. So, we only keep the *in vitro* IFN- γ production in CD8 T cells to further demonstrate the immunosuppressive mMDSC could inhibit the proliferation and IFN- γ production of antiviral CD8 T cells *in vitro* in Fig. 4g to 4l. Additionally, we found there were significantly lower number of T cells infiltrated in heart from Trim29^{MyHC-KO} mice in comparison to Trim29^{fl/fl} in Supplementary Fig. 6c. We speculate that is why extent of inflammation shown in Fig. 3 by H&E staining and cytokine levels was significantly lower in Trim29^{MyHC-KO} mice, even if CD8 T cells produced more IFN- γ in heart from Trim29^{MyHC-KO} mice.

10. The authors showed only a gating strategy for the spleen but not for the heart in Extended Data Fig. 5. They should present a gating strategy for the heart.

Reply: Great suggestion. We have provided the gating strategy for the heart in Supplementary Fig. 6d.

11. This manuscript is readable, but it is strongly recommended to have it proofread by a native English speaker to improve clarity.

Reply: This is a great point. We have it proofread by a native English speaker as shown in Acknowledgements section.

Reviewer #2 (Remarks to the Author):

Thanks for the revised manuscript. It is much improved!

Reviewer #3 (Remarks to the Author):

The Authors properly addressed the comment raised.

I only have a minor comment on the clarity of the Graphs' labelling of Fig. 1, 2, and 4. The open versus full circles to indicate TRIM29 KO and WT, respectively, is confusing with the dark and white bars of the graph. In some cases the 3 circles are on the edge of the bar rendering difficult and non immediate the interpretation.

Point-by-point reply

REVIEWER COMMENTS

Reviewer #2 (Remarks to the Author):

Thanks for the revised manuscript. It is much improved!

Reply: We are grateful to the reviewer for dedicating time to thoroughly evaluate our manuscript and for providing exceptionally insightful and constructive feedback.

Reviewer #3 (Remarks to the Author):

The Authors properly addressed the comment raised.

I only have a minor comment on the clarity of the Graphs' labelling of Fig. 1, 2, and 4. The open versus full circles to indicate TRIM29 KO and WT, respectively, is confusing with the dark and white bars of the graph. In some cases the 3 circles are on the edge of the bar rendering difficult and non immediate the interpretation.

Reply: We greatly appreciate the time and effort you have invested in reviewing our manuscript and offering comprehensive feedback. Your insightful suggestions regarding the clarity of our graph labels have been invaluable.

To address your recommendations and prevent any potential confusion for readers, we have switched the dark and white colors of the bar graphs between Trim29^{+/+} and Trim29^{-/-} (in Figs. 1c-1g, 1i and Figs. 2a,2f), Trim29^{fl/fl} and Trim29^{MyHC-KO} (in Figs. 4b,4d,4h,4j,4l), sh-Ctrl and sh-T29 (in Supplementary Figs. 1d-1h,1j and Figs. 2a,2f), WTMyHC-Cre and Trim29^{MyHC-KO} (in Supplementary Figs. 6a,6b) or Trim29^{fl/fl} and Trim29^{MyHC-KO} (in Supplementary Fig. 6c), so that the white and black open versus full circles (or triangles) matches perfectly with the revised dark and white colors of the bar graphs now.

We trust that these revisions will enhance the figure legibility and greatly appreciate your guidance in improving the presentation of our data.